# Modelling chemotaxis of branched cells in complex environments provides insights into immune cell navigation

Jiayi Liu[1,2], Jonathan E. Ron[2], Giulia Rinaldi[3], Ivanna Williantarra[3], Antonios Georgantzoglou[3,4], Ingrid de Vries[5], Michael Sixt[5], Milka Sarris[3], Nir S. Gov[2,3]*

1 Department of Physics, Yale University, New Haven, Connecticut, United States of America, 2 Department of Chemical and Biological Physics, Weizmann Institute of Science, Rehovot, Israel, 3 Department of Physiology, Development and Neuroscience, Downing Site, University of Cambridge, Cambridge, United Kingdom, 4 Novo Nordisk Foundation Center for Stem Cell Medicine (reNEW), Department of Biomedical Sciences, University of Copenhagen, Copenhagen, Denmark, 5 Institute of Science and Technology Austria (ISTA), Klosterneuburg, Austria

* nir.gov@weizmann.ac.il

## Abstract

Cell migration *in vivo* is often guided by chemical signaling, i.e., chemotaxis. For immune cells performing chemotaxis in the organism, this process is influenced by the complex geometry of the tissue environment. In this study, we use a theoretical model of branched cell migration on a network to explore the cellular response to chemical gradients. The model predicts the response of a branched cell to a chemical gradient: how the cell reorients its internal polarity and how it navigates through a complex environment up a chemical gradient. We then compare the model's predictions with experimental observations of neutrophils migrating to the site of a laser-inflicted wound in a zebrafish larva fin, and neutrophils migrating *in vitro* inside a regular lattice of pillars. We find that the model captures the details of the subcellular response to the chemokine gradient, as well as qualitative characteristics of the large-scale migration, suggesting that the neutrophils behave as fast cells, which explains the functionality of these immune cells.

## Author summary

Cells often migrate through complex tissue environments by following chemical signals, a process known as chemotaxis. Immune cells like neutrophils, must rapidly navigate through complex tissue structures to reach sites of injury or infection. While migrating, these immune cells become highly branched, with multiple protrusions extending in-between the surrounding tissue cells. Here, we develop a theoretical model that describes how branched cells migrate in a network in response to chemical gradients. The model reveals a trade-off between speed and accuracy under low internal cellular noise: slower cells follow weak gradients more

**Data availability statement:** The simulation code used to generate all computational results in this study is provided in the Supporting information (S1 Code). Parameter values and analysis settings are specified in the Methods and figure captions.

**Funding:** N.S.G. is the incumbent of the Lee and William Abramowitz Professorial Chair of Biophysics (Weizmann Institute), and acknowledges support from the Royal Society Wolfson Visiting Fellowship, and Human Frontier Science Program grant RGP0032/2022. Work by M.S., I.W., G.R. and A.G. was supported by the Leverhulme Trust (grant RPG-2021-226) and the European Research Council (ERC) under the Horizon 2020 program and UKRI, Grant agreement No. EP/Y02799X/1. M.S. and I.d.V acknowledge support by the European Research Council (grant ERC-SyG 101071793 to M.S). The funders had no role in study design, data collection and analysis, decision to publish, or preparation of the manuscript.

**Competing interests:** The authors have declared that no competing interests exist.

precisely, whereas faster cells reach the target more rapidly but with reduced directional accuracy. However, this relationship becomes much less pronounced in the presence of realistic levels of noise. We validate our model by comparing it to experiments of neutrophils migrating to a wound in zebrafish tissue and within a controlled lattice of micro-pillars. The data suggest that neutrophils operate in a fast regime, enabling rapid responses at the cost of accuracy in weak gradients. This study offers new insights into how immune cells respond to chemotactic gradients in complex environments, and provides a novel theoretical framework for understanding branched cell chemotaxis *in vivo*.

## Introduction

Tissues have complex geometries and present a great diversity of topologies which force embedded motile cells to deform and exhibit a plethora of branched shapes [1], as they form new extensions around the surrounding obstacles [2]. Prototypical examples include immune cells that continuously scan the tissues in search for signals and need to migrate within the complex tissues towards sites of wound and infection [3–5], or cancer cells that spread from the main tumor [6]. In order to efficiently migrate towards their target, immune cells must coordinate their branching dynamics to navigate the microenvironment and decide on the new migration direction, in a process we refer to as directional decision-making (DDM) [7,8]. Unlike mesenchymal modes of movement, whereby special adhesive interactions with extracellular matrix condition cellular guidance, immune cells navigate faster and with relative flexibility in tissues, in a mode often referred to as amoeboid locomotion [9, 10]. Despite its clear biological importance, the mechanisms of amoeboid branched cell migration and chemotaxis in complex tissue geometries remain poorly understood [11], even for single-cell chemotaxis where collective effects are less important [12,13]. Furthermore, there is currently a lack of theoretical framework for tackling these questions.

Chemotaxis of cells, i.e. the ability to interpret chemoattractant gradients, is crucial for navigation to sites of injury or infection. This process has been associated with the ability of cells to grow protrusions in different directions to better sample their surroundings and the guiding chemical gradient. This process has been largely studied for cells migrating unhindered on flat surfaces, where they migrate, while spontaneously growing new branches, mostly at their leading edge, at different angles with respect to the direction of motion and the chemical gradient [14–16]. Chemotaxis was also studied for cells migrating within confined channels and through junctions [17–21], including barotaxis [22]. The theoretical treatment of the chemotaxis of spontaneously branching cells [23–25] has generated insights regarding the relationship between cell shape dynamics and accuracy during chemotaxis. However, many of these models are very complex due to the need to describe both the cell shape, polarization, and spontaneous branching processes (at different angles). Alternatively, some of these models maintain only schematic description of the cell shape [25] and internal polarity field [22], which prevents the description

of the intricate shape dynamics observed when cells migrate over junctions [26]. It therefore remains as an open problem to model and understand the chemotaxis of cells within networks where the branching process is enforced by the confining geometry, which is the focus of this paper.

To address this challenge, we have recently developed a theoretical model describing the shape dynamics of branched migrating cells performing DDM over single junctions [26], and the spontaneous polarization and migration of highly branched cells on hexagonal networks [27]. The coarse-grained model describes cells migrating on an hexagonal network, composed of linear segments (Fig 1C), with an internal mechanism for spontaneous cellular polarization and migration (see Theoretical model for details). This minimalist model therefore provides us with a framework to study chemotaxis of highly branched migrating cells, where the branching is determined by the network geometry.

Here we utilize this theoretical model to explore the chemotaxis characteristics of branched cells. To test the predictions from the simulations, we compared them to two systems (Fig 1), one *in vivo* using neutrophil responses to tissue injury in zebrafish [5,29] and one *in vitro* where neutrophils migrate in a regular lattice of pillars [28]. Neutrophils are fast migratory leukocytes able to rapidly enter inflamed tissues, adopt cell shapes and respond to a range of chemical gradients generated by tissue-cells or themselves. Thus, the two experimental systems used here represent biologically relevant situations where cells must negotiate complex geometries with DDM in response to chemical gradients.

We validate the model by comparing it to the experimental data of the shape dynamics and migration characteristics of neutrophils as they respond to a signal that recruits them to a wound, in the zebrafish skin [5]. Our model reveals a speed–accuracy trade-off under low internal cellular noise, whereby slow cells are accurate, while fast cells make meandering paths in weak chemical gradients (far from the chemical source). However, this effect is diminished when the internal cellular noise is at a realistic level. In addition, the model predicts that fast cells undergo distinct events where

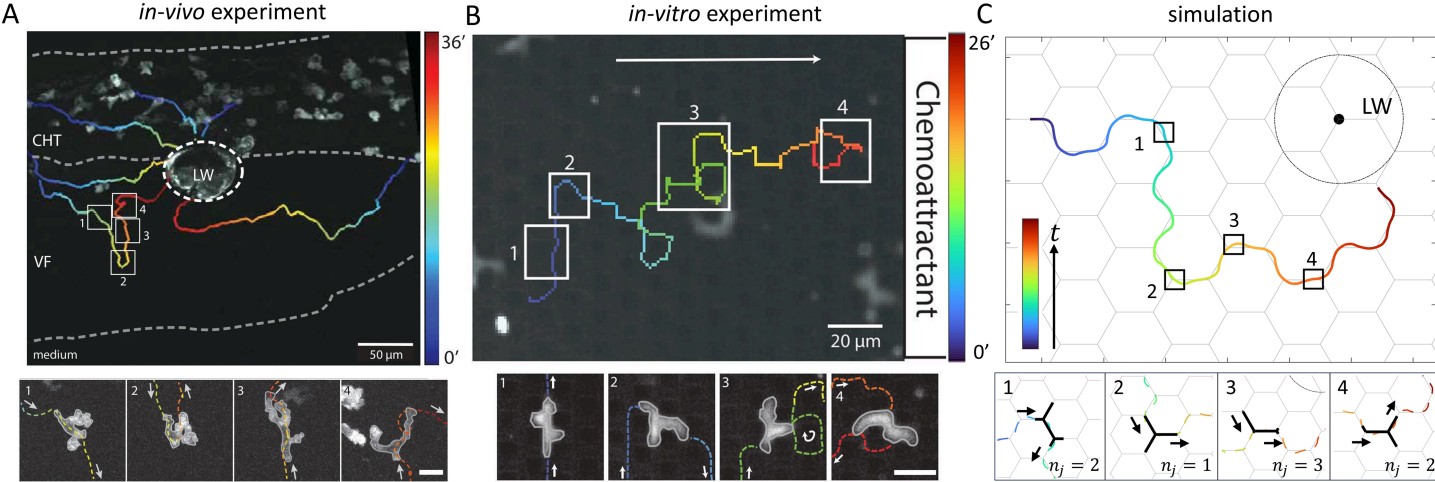

**Fig 1**. **Branched cell migration in complex environments:** *in vivo*, *in vitro*, **and simulated trajectories with shape dynamics.** (A) Morphological dynamics of neutrophil swarming during migration from the caudal hematopoietic tissue (CHT) toward a laser-inflicted wound (LW) at the ventral fin (VF)–CHT boundary in transgenic zebrafish larvae expressing the calcium indicator GCamp6f (see *In vivo* experimental methods for details). Top panel: Representative trajectories of neutrophils migrating toward the LW, with colors denoting migration time. White boxes highlight time points along the trajectory of an individual cell, corresponding to the snapshots in the bottom panels (1–4). White arrows indicate migration direction. Scale bars: 50 μm for trajectories and 10 μm for snapshots. (B) Trajectory and snapshots of a cell migrating by chemotaxis within a regular pillar lattice (see *In vitro* experimental methods for details) [28]. (C) Simulation of a branched cell migrating toward a chemokine point source. Top panel: Representative trajectory of the cell's center-of-mass (C.O.M.), with colors denoting migration time. Black boxes mark time points along the trajectory, corresponding to the snapshots in the bottom panels (1–4). Black arrows indicate migration direction. Key parameters: $C/c_0 = 0.01, \epsilon = 0.2, d = 3, \beta_0 = 8, \sigma = 0.8$.

they transiently get trapped in indecision, with competing protrusions pulling the cell in a tug-of-war around the obstacle. Comparing these characteristics predicted by the model to the experiments (both *in vivo* and *in vitro*), we find that neutrophils behave as fast cells in our model, thereby providing insights into the constraints that determine the optimal properties for efficient neutrophils response.

## Materials and methods

### Ethics statement

The experiments in this study involved only established human cell lines and did not include any human participants or animal subjects. Therefore, no ethics approval was required.

### Theoretical model

We provide here the theoretical model describing the shape dynamics of highly branched cells [27], which is an extension of our model for cells migrating over a single junction [26].

**Model equations.** For a cell with $N$ ($N \geq 3$) arms, i.e., spanning across $N$–2 junctions (Fig 2), the dynamics of the arm $i$ are described by three variables: 1) its length $x_i$, 2) the concentration of the adhesion bonds $n_i$ at the leading edge, and 3) the local actin treadmilling flow velocity $v_i$ at the leading edge. The dynamical equations for these variables are given by [27]

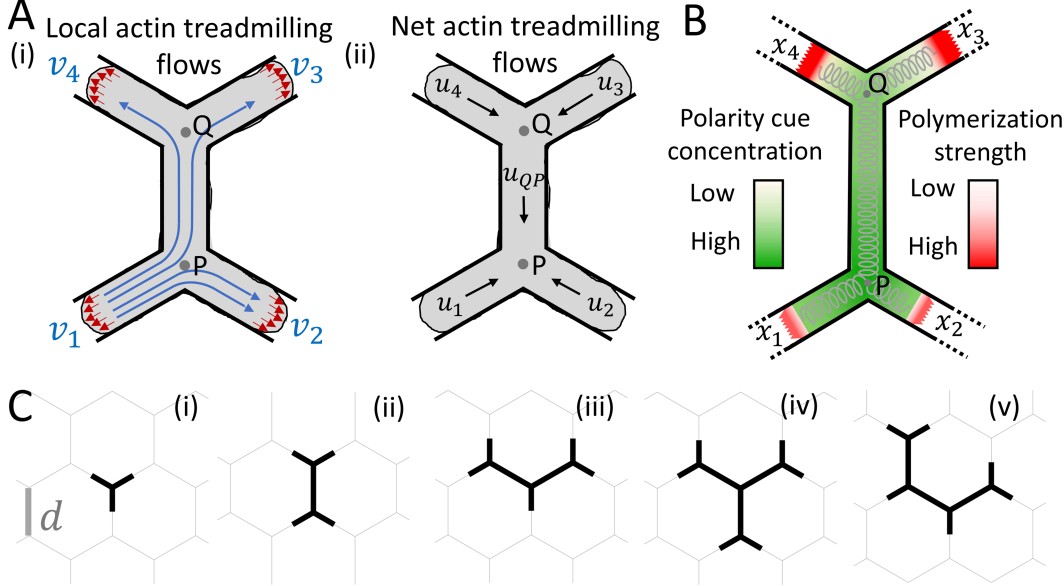

**Fig 2. Theoretical model of branched cell polarization.** (A) (i) Schematic of local actin treadmilling flows $v_i$ at the arm edges, with the symmetric splitting of the flow from arm 1 at each junction. Red arrows denote the protrusive force proportional to the local flows. (ii) Schematic of the net actin treadmilling flows within each cell segment. (B) Schematic of the polarity cue concentration field which is affected by the advection flows (green scale), and the corresponding polymerization strength (red scale). Cell elasticity is represented by gray springs. (C) Simulation snapshots of cells spanning different numbers of junctions: (i) one, (ii) two, and (iii) three junctions, and (iv–v) two possible shapes for cells spanning four junctions.

$$\dot{x}_i = \frac{1}{\Gamma_i}[v_i - k(L-1)] \tag{1}$$

$$\dot{n}_i = r(1 - n_i) - n_i \exp\left[\frac{-v_i + k(L-1)}{f_s n_i}\right] \tag{2}$$

$$\dot{v}_i = -\delta(v_i - v_i^*) + \sigma\xi_t \tag{3}$$

Eq 1 is a simplified description of the protrusive traction forces, which can be further elaborated to include the adhesion dependence of these forces [30]. In our model, we assume that the protrusive force exerted on the membrane is proportional to the local actin treadmilling flow $v_i$, with the proportionality constant normalized to unity [26] (Fig 2A(i)).

In Eq 1, $\Gamma_i$ is a non-constant friction coefficient that depends on the direction of motion of the arm's leading edge, given by

$$\Gamma_i = \begin{cases} 1, & \dot{x}_i > 0, \\ n_i\kappa \exp\left(\dfrac{-v_i + k(L-1)}{f_s n_i}\right), & \dot{x}_i < 0, \end{cases} \tag{4}$$

where $\kappa$ is the effective elasticity of the adhesion bonds. The friction acts as a constant drag $\Gamma_i = 1$ when the arm is extending, while during retraction, the friction depends explicitly on the adhesion level $n_i$ governed by the slip-bond adhesion dynamics in Eq 2. Therefore, adhesion dynamics directly modulates the retraction rate in Eq 1.

The restoring force of the cell elasticity in Eqs 1, 2 is described by a simple spring term $k(L-1)$, where $k$ is the effective elasticity of the cell (of rest length 1), and the total length of the cell is

$$L = \sum_i x_i + (N-3)d \tag{5}$$

where $d$ is the grid size, i.e., the distance between two adjacent junctions on the network. Note that the effective spring constant $k$ describes here the overall restoring force of the cell, including its material elasticity and active contractility components.

Eq 2 describes the dynamics of the slip-bond adhesion molecules that give rise to the stick-slip migration. In Eq 2, $f_s$ describes the susceptibility of the slip-bonds to detach due to the applied force which is shared equally among $n_i$ bonds, and $r$ is the effective cell-substrate adhesiveness. The friction and adhesion dynamics which we have in our model (Eqs 2, 4) give rise to the stick-slip migration mode. The stick-slip behavior is important for the dynamics of the branched cell (see Results section), but we emphasize that it is not the only driving mechanism for oscillations and non-linear dynamics. The non-linear coupling between the polarity-cue distribution and the actin treadmilling flow (S1 Appendix) gives rise to sharp transitions in the polarity between competing arms, and is the main mechanism driving oscillations in this dynamical system [26,27].

In Eq 3, $\delta$ is the rate at which the local actin flows converge to the steady-state solutions $v_i^*$. The term $\sigma\xi_t$ in Eq 3 describes the internal noise in the actin treadmilling flows, modeled as a Gaussian noise with amplitude $\sigma$.

In our model, the net actin flow advects a polarity cue that acts as an inhibitor of local actin treadmilling at the arm tips (Fig 2B). The steady-state local flow at the tip of arm $i$ is given by

$$v_i^* = \beta\frac{1}{1 + \bar{c}_i(x_i)} \tag{6}$$

where $\beta$ is the maximal actin polymerization speed at the arm tips (hereafter referred to as "actin activity"), and $\tilde{c}_i(x_i)$ is the normalized concentration of the inhibitory polarity cue at the tip of arm $i$, which incorporates contributions from the net actin flows within all cell segments. The detailed calculation of $\tilde{c}_i(x_i)$ and $v_i^*$ is provided in S1 Appendix.

The net actin flow inside arm $i$ (Fig 2A) is given by

$$u_i = v_i - \sum_{j \neq i} \frac{v_j}{2^{m_{i,j}}} \tag{7}$$

where $m_{i,j}$ is the number of junctions between arm $i$ and arm $j$. Note that the relation $\sum_{j \neq i} \frac{1}{2^{m_{i,j}}} = 1$ always holds.

The net actin flow inside a node-connecting segment that connects two adjacent junctions Q and P (Fig 2A) is given by

$$u_{QP} = \sum_{i\ connect\ to\ Q} \frac{v_i}{2} - \sum_{j\ connect\ to\ P} \frac{v_j}{2} \tag{8}$$

where $i$ and $j$ denote the free arms that emanate from junction Q and P, respectively.

**Cell polarization.** In our model, cell polarization is defined as the spontaneous symmetry breaking among all arms, whereby a non-zero net actin flow $u_i$ (Eq 7) emerges as a self-consistent solution of the model equations. This transition from non-polar (zero net actin flow) to polar state strongly depends on the total cell length [31]. For a symmetrically spreading cell, the critical length $L_p$ at which the actin protrusive force balances the cell elasticity, is given by

$$L_p = \frac{1}{2}(1 - c) + \frac{\beta}{2k} + \sqrt{c + (\frac{1-c}{2} + \frac{\beta}{2k})^2} \tag{9}$$

Another critical length is $L_c$, above which the actin flows in the arms deviate from the uniform solutions. For the single-junction case, it is given by

$$L_c = \frac{c}{\sqrt{c\beta/2D - 1}}. \tag{10}$$

For the multiple-junction case, $L_c$ also depends on the grid size $d$ [27].

By equating Eq 9 and Eq 10, we obtain the critical actin activity $\beta_c$ for cells to polarize on a single junction

$$\beta_c = \frac{D}{c} + ck + \frac{ck^2}{4D} + \frac{2D - ck}{4cD}\sqrt{4D^2 + 4c(1 + 2c)Dk + c^2k^2} \tag{11}$$

When $\beta < \beta_c$ ($L_p < L_c$), the cell spreads symmetrically and remains stationary at the junction(s) with a total length of $L_p$. In contrast, when $\beta > \beta_c$ ($L_p > L_c$), the cell continues spreading until its length exceeds $L_c$, at which point it spontaneously polarizes and migrates along one of the arms.

For cells spanning multiple junctions, to determine the critical $\beta$ for migration, we further consider the minimum cell length required for a cell with $N>3$ arms (i.e., spanning $N–2$ junctions)

$$L_{min} = (N - 3)d \tag{12}$$

By equating Eq 9 and Eq 12, we derive the critical actin activity for cells to span $N–2$ junctions, $\beta_d$. By further equating Eq 9 and Eq 10, under the condition that the cell spans at least $L_{min}$ (Eq 12), we obtain the critical actin activity for cell

migration on multiple junctions, denoted as $\beta_c$. In general, $\beta_c$ increases with $N$ within the parameter ranges used in this study. Details of these calculations can be found in [27].

## Numerical implementation and data analysis

All numerical simulations based on the theoretical model were performed using custom-written codes in Fortran 90 (S1 Code). Eqs 1, 2 and 3 were integrated numerically using the Euler-Maruyama method with a fixed time step, following the implementation described in [26]. Post-processing, statistical analysis, and visualization of the simulation results were carried out in MATLAB R2024a (MathWorks).

### *In vivo* experimental methods

**Zebrafish Husbandry and preparation.** Tg(lyzC:Gcamp6f) transgenic zebrafish line were maintained adhering to the UK Home Office regulations, UK Animals (Scientific Procedures) Act 1986, which was reviewed by the University Biomedical Service Committee. Adult zebrafish were maintained and bred according to standard protocols [32]. Embryos were collected at 3 hours post fertilisation, bleached for 5 minutes using 0.003% NaOCl (Cleanline, CL3013), followed by three times washing in E3 media (as described in [33]). Embryos were then raised at 28 °C in E3 medium supplemented with 0.003% PTU (1-phenyl 2-thiourea; Sigma Aldrich, P7629) to inhibit pigmentation and maintain optical transparency. 3 days post-fertilization (dpf) larvae were used on the day of imaging.

**Bacterial culture and preparation.** A day before the imaging day, a single colony of *Pseudomonas aeruginosa* (strain PAO1 from Martin Welch), isolated via a four-way streak on Pseudomonas Isolation Agar (BD Difco, 292710) supplemented with cetrimide and nalidixic acid (E&O Laboratories Ltd, LS0006) and glycerol (Fisher Scientific, G/0650/17), was grown in 5 mL of antibiotic-free LB broth (Formedium, LBX0102) at 37 °C with shaking for 24 hours. On the imaging day, the overnight culture was diluted and incubated to logarithmic phase ($OD_{600}$ = 0.6–0.8). Bacterial concentration was adjusted to $3 \times 10^5$ CFU/mL using spectrophotometric estimation, based on the assumption that $OD_{600}$ = 0.5 corresponds to $1 \times 10^7$ CFU/mL. Bacteria were pelleted and washed twice with PBS (Sigma Aldrich, D8537) to remove residual medium before resuspension in Ringer solution (as described in [29]) containing $0.16 \, \text{mg mL}^{-1}$ (1x) Tricaine (Merck, A5040).

**Zebrafish mounting and infection.** Transgenic larvae expressing the most prominent calcium indicator, Gcamp6f (cDNA originally described by [34]), were screened to ensure high expression. Larvae displaying strong fluorescence and healthy morphology were selected for imaging. Anesthetised larvae were mounted in a 1:1 mix of 2% low-melting agarose (Invitrogen, 16520) and 2× Tricaine in a custom-built coverslip chamber, composed of glass coverslips covering both the top and bottom of the samples and sealed onto a metallic ring. Larvae were oriented laterally and agarose was allowed to solidify. After the agarose was solidified, the tail fin was exposed by carefully removing the surrounding agarose under a dissecting microscope using a tweezer and a capillary needle, allowing bacterial access post-wounding. The chamber was then filled with 1 mL of Ringer solution containing 1x Tricaine and the diluted PAO1 suspension before sealing.

**Two-photon imaging and laser wounding.** Laser ablation [29] and time-lapse imaging were performed using a LaVision TriM Scope multiphoton microscope equipped with an electro-optic modulator for rapid power modulation. A Spectra-Physics Insight DeepSee dual-line laser was tuned to 900 nm for imaging and 1040 nm for ablation, with the imaging power set to approximately 500 mW at the specimen plane. Image acquisition was performed using ImSpector Pro software (5.0.284.0, LaVision Biotec). Two-photon microscopy was performed using a 25×/NA 1.05 water dipping objective lens, with $ddH_2O$ applied to the lens and the correction collar set to 0.17 to match the refractive index. Out of all the mounted larvae, the larvae with the best overall health (normal circulation, heartbeat and morphology) and no damages or aberrant wounding due to mounting was selected for image acquisition. The ventral fin within the caudal hematopoietic tissue of the selected larvae was then centered using brightfield optics. A region of interest with a diameter of 40 μm was defined as the wound area on a single focal plane (as superficial as possible), scanned at 240 nm/pixel with a 15 μs dwell time. Z-stacks were acquired every 20 s with a step size of 2 μm, consisting of ~20 planes. Laser wound

was triggered after a 2-minute pre-wound baseline, followed by 3 hours of post-wound imaging. Focus was maintained throughout acquisition. Image stacks were processed in Fiji (ImageJ 1.52p, June 2019, publicly available; [35]) using maximum intensity z-projections to generate neutrophil swarming timelapse videos of which were then used for subsequent analysis.

**Image analysis.** Analysis of neutrophil trajectories was performed in Imaris v8.2 (Bitplane AG) on 2D maximum intensity projections of the 4D time-lapse movies (*in vivo* experiments) and 3D time-lapse movies (*in vitro* experiments) (as previously described in [5,33]). Cell trajectories were manually tracked overtime (Figs 1 and 7), speed and straightness coefficient of the trajectories were calculated with the imaging software. The projected area of the cells overtime was calculated by manually drawing the perimeter of the cells in Fiji (ImageJ) and using the measure tool.

### *In vitro* experimental methods

We describe here the microfabrication process of the PDMS devices, their design and manufacturing (S1 Fig) [28].

**Photomask.** The Photomask is chrome patterned on a glass plate and used to fabricate the wafer. Photomask design is drawn in CorelDraw X8 and exported to DXF file format. LinkCad is used to convert DXF to Gerber format. Photomasks are manufactured by https://www.jd-photodata.co.uk/

**Wafer.** The wafer was baked for 5 minutes at 110 °C, followed by spin-coating SU8 6005 TF (MICROCHEM) and prebaking for 5–10 minutes at 110 °C. The wafer was exposed to 100 mJ cm$^{-2}$ and post-baked at 110 °C for 5 minutes. This was followed by a developing step with SU8 developer and IPA, and final baking at 135 °C for 5 minutes. The height of the device was measured using a profilometer. Finally, the wafer was silanized with 1H,1H,2H,2H perfluorooctyltrichlorosilane for 1 hour in a sealed vacuum.

**PDMS devices.** 40 mL 1:10 PDMS Sylgard 184 (Dow Corning) was mixed and degassed for 2 minutes at 2000 rpm and for 2 minutes at 2200 rpm in a Thinky ARE-250 mixer/defoamer, then poured on the wafer in an aluminum mold, degassed in a desiccator, and cured for 2 hours at 80 °C. The devices were cut into small squares and 2.5 mm holes were punched (Harris Unicore biopsy puncher) on both sides as loading ports. The devices were cleaned with tape (Scotch Magic Tape) and air blown. Cover slip (#1.5, 22 × 22 mm, Menzel) and device were plasma activated for 2 minutes in a plasma cleaner (Harrick Plasma). The activated side of the device was placed on the charged side of the coverslip and baked at 95 °C for 15 minutes to achieve bonding. Devices attached to the coverslips were glued with Paraplast X-tra (Sigma) onto the bottom of a 6 cm cell culture dish so that it covers a central hole of 17 mm diameter. 1–2 hours before adding cells, devices were incubated with R10 media (RPMI 1640 media [21875091, Gibco] supplemented with 10% FCS [Gibco] and Penicillin-Streptomycin), in a cell culture incubator at 37 °C and 5% $CO_2$.

**PLB-985 (promyelocytic leukemia blasts) cells.** PLB-985 cells (promyelocytic leukemia blasts) were obtained from the DSMZ (PLB-985 ACC 139). Cells were grown in R10 media. Three to four days before the experiment, cells were differentiated by adding 1.25% DMSO to the R10 media. Before the experiment, cells were stained with 10 µm TAMRA (Invitrogen) diluted in PBS for 5 minutes in the dark and washed 3× in R10 media.

Imaging cells in micropillar devices: liquid was removed from both loading ports. One port was filled with 5–7 µL of fMLP (50 mmol L$^{-1}$) and the other port with 5–7 µL of cell suspension (20,000 cells/µL). The loaded devices were placed in the incubator for a minimum of 1 hour. Movies of cells migrating in the devices were acquired with an imaging interval of 10 s using a Nikon Ti2E inverted widefield microscope equipped with a Plan Apo $\lambda$ 20×/0.75 DIC 1 air PFS objective, a monochrome CMOS sensor camera, and a custom-built climate chamber (37 °C, 5% $CO_2$, humidified).

## Results

### Chemotaxis over a single junction

We begin by studying chemotaxis of a cell migrating over a single junction. We extend our previous model (see Theoretical model) [26,31] by introducing a fixed external chemical signal. This simple description of chemotaxis neglects the

feedback from the cells on the chemical gradient [12,13], which is important to consider when cells are at high densities inside highly confined geometries. Chemokine binding to the receptors at the leading edge locally enhances the activity of the actin polymerization machinery [36]. In our model, this is represented by increasing the actin activity parameter $\beta$ at the tip of the arm that faces the source (arm 3, Fig 3A)[37]:

$$\beta = \beta_0 (1 + \epsilon) \tag{13}$$

where $\beta_0$ is the baseline actin activity and $\epsilon$ is the relative enhancement due to the signal. At the tips of arms 1 and arm 2, the actin activity remains unchanged with $\beta = \beta_0$. This simulation setup is similar to chemotaxis experiments in cells [37], and the localized actin polymerization photo-activation used to direct cells over junctions [19].

As in our previous work [26], the model equations (Theoretical model) are normalized by the inverse focal-adhesion disassembly timescale (5–30 min [38–41]) and by the cell's rest length scale (10–100 μm on one-dimensional tracks [31]), so the model parameters are dimensionless. Throughout, we use the same parameter set calibrated to typical motile cells [26] (Table A in S1 Appendix). Our aim is to explore the general properties of branched cell chemotaxis in complex networks, independent of specific values of the model parameters.

Our main control parameter is the baseline actin activity $\beta_0$. It is bounded below by $\beta_c$, the minimal value that allows the polarization [31] and migration over the network [26]. This value is weakly dependent on the cell shape, increasing with the number of network nodes and arms that the cell has. We therefore explore $\beta_0 > \beta_c$ ($\beta_c \sim 5-6$ for the model parameters chosen here, calibrated in [26]). Above a higher threshold $\beta_c^{slow}$, the model predicts a "slow mode" in which two leading arms become highly elongated while the rear remains stuck at the same node, until the competition between the elongated arms is resolved [26]. For our parameters, $\beta_c^{slow} \sim 9-12$. These considerations determine the range of $\beta_0$ explored in this paper (see Fig 3B as an example).

In Fig 3B(i,ii), for a fixed small bias $\epsilon = 0.001$ (Eq 13) and two levels of cellular internal noise in the actin treadmilling flows (Eq 3), we plot the mean escape time $\langle T_{esc} \rangle$ and the error rate of decision-making $P(wrong)$ versus $\beta_0$. Here, a "wrong" decision means exiting along arm 2 or arm 1 (rather than arm 3 toward the source). The reflection probability $P(back)$ (blue) quantifies turns back and exits along arm 1. For cells that did not make a decision within the maximal simulation time $T_{max} = 100$, we assigned the direction of the longest arm at the end of the simulation as the cell's eventual migration direction. As expected, enhancing the actin activity at the arm facing the chemoattractant biases decisions towards that arm. This behavior is consistent with experiments on spontaneously branching cells, where the branch that points up the chemical gradient has a higher probability of becoming the new leading edge of the cell [14].

The region where the cells perform stick-slip migration over the junction (blue shading) coincides with nonzero $P(back)$. The onset of the stick-slip motion is discussed in S2 Appendix. Representative dynamics during different cases of DDM are shown in Fig 3C-E, for the three $\beta_0$ values marked by black dashed lines in Fig 3B(i). During both stick-slip and slow-mode events, the simulated cell length changes by a factor of ~2 (Fig 3D and 3E), which is similar to the length variations observed in experiments (where cell area serves as proxy for its total length) (Fig 7A,B).

For a weak noise ($\sigma = 0.1$, Fig 3B(i)), $\langle T_{esc} \rangle$ initially decreases with $\beta_0$, which is expected as the cell moves faster at higher actin activity [26]. $\langle T_{esc} \rangle$ rises sharply for $\beta_0 \sim 10$, as cells tend to get stuck in the slow-mode event for a long time, with two elongated and competing arms (Fig 3E), consistent with the no-chemotaxis case [26]. In this mode, the rear arm (arm 1) shrinks nearly to zero, while the two new arms remain similarly long until one of them retracts rapidly.

The accuracy, quantified by $P(wrong)$, shows the opposite trend, exhibiting a "speed–accuracy trade-off": cells that spend less time over the junction are more likely to make incorrect decisions and miss their way to the target. The abrupt rise of $P(wrong)$ near $\beta_0 \sim 8$ reflects a finite reflection probability (blue line in Fig 3B(i)) that appears only in stick–slip regime [26,31] (blue shading in Fig 3B(i)). Examples of the smooth and stick-slip migrations are shown in Fig 3C and D, respectively. For smooth motion, the only incorrect decision is exiting along arm 2. However, during stick-slip events, the total cell length falls below the critical polarization length $L_c$ (horizontal dashed line in Fig 3D; Theoretical model), leading

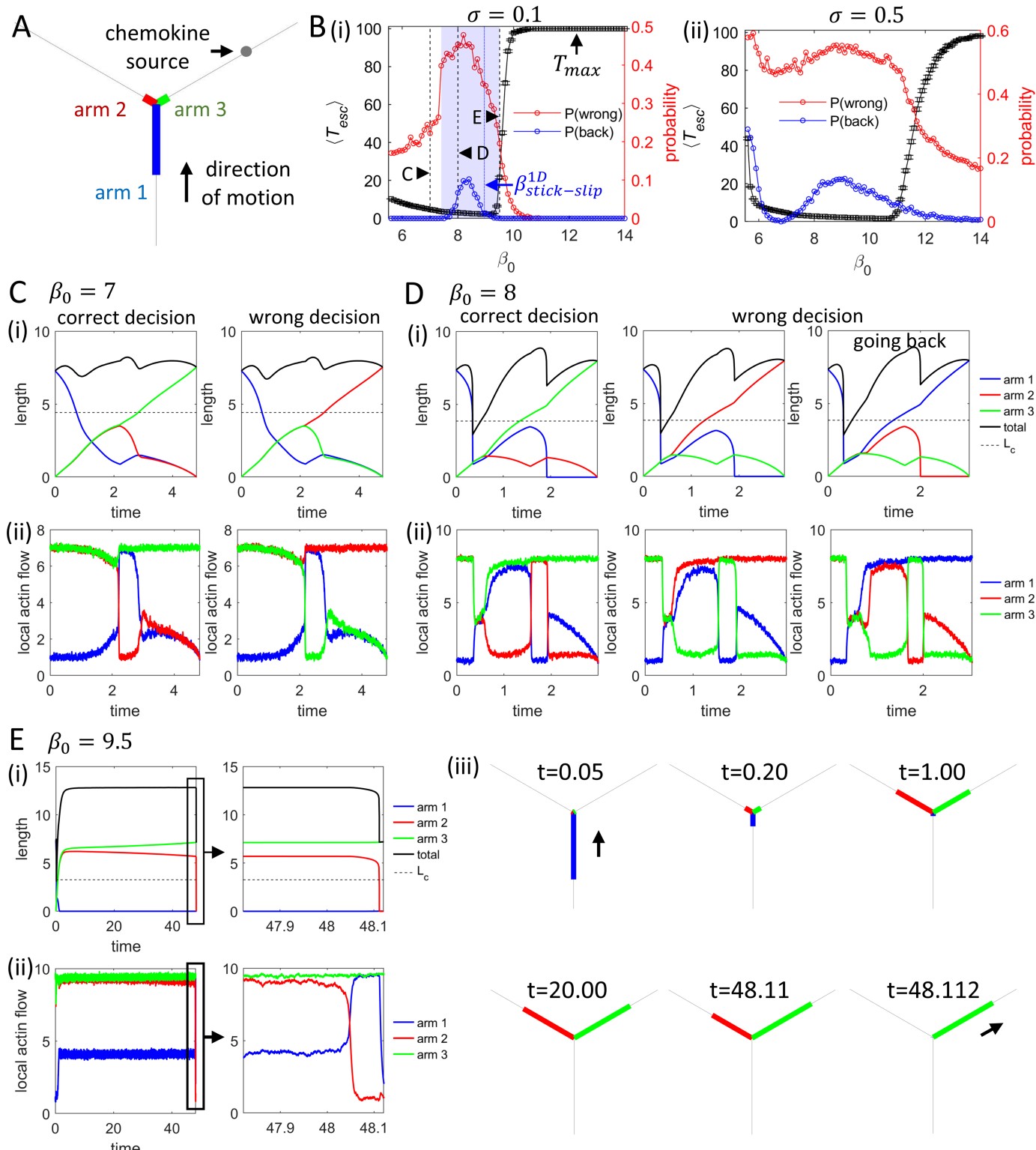

**Fig 3. Chemotaxis dynamics of cells at a single junction with a localized chemokine point source.** (A) Schematic of the model where the chemokine source is confined to one of the new arms that extend from the junction (with respect to the direction that the cell arrives at the junction). (B) Mean escape time across the junction, $\langle T_{esc} \rangle$ (black), error rate $P(wrong)$ (probability of leaving along arm 2 or arm 1, red), and the probability of being "reflected" at the junction $P(back)$ (probability of leaving along arm 1, blue), as functions of $\beta_0$. Maximal simulation time: $T_{max} = 100$. The blue

shaded region in (i) marks the range where cells undergo stick-slip events, allowing a cell to reverse its path ($P(back)>0$). The vertical blue line in (i) indicates the threshold value of $\beta$ for stick-slip migration with the same parameters on an infinite line (S2 Appendix), which is larger than the value needed to initiate stick-slip by the arrival of the cell at the junction. Panels (i) and (ii) correspond to two levels of the internal noise in the actin activity, $\sigma = 0.1$ and $\sigma = 0.5$, respectively. In both B(i) we find a speed–accuracy relation for the cell's chemotaxis: faster migration over the junction (low $\langle T_{esc} \rangle$, black) results in larger error ($P(wrong)$, red). However this effect is almost washed out for higher noise (B(ii)). 2000 stochastic simulations were performed to obtain the statistics. Error bars represent mean $\pm$ standard error. (C-E) Dynamics of (i) arm lengths and total cell length, as well as (ii) local actin flows at the arm tips, for representative events: correct decision (exit via arm 3), wrong decision moving forward (exit via arm 2), and wrong decision by turning back (exit via arm 1). Panel E(iii) shows snapshots of the simulated cell corresponding to E(i,ii). (C), (D) and (E) correspond to $\beta_0 = 7.0$, $\beta_0 = 8.0$ and $\beta_0 = 9.5$, respectively (as indicated by the vertical dashed lines in B(i)). Other key parameter: $\epsilon = 0.001$.

to polarity loss and allowing exits back along arm 1. This drives the sharp increase in errors, similar to the behavior at high noise and very low $\beta_0$, as discussed below.

As $\beta_0$ increases further, cells enter the slow mode regime. The slow-mode probability $P(slow)$ as a function of $\beta_0$ (black line in S2A Fig) closely follows the $\langle T_{esc} \rangle$ trend (black line in Fig 3B(i)). Moreover, the error rates in the fast and slow modes (red and green lines in S2 FigA, respectively) show that the slow mode is significantly more accurate.

Finally, for selected $\beta_0$ values (black dashed lines in S2A Fig), S2B Fig shows the distributions of $T_{esc}$ (histograms) and $P(wrong|T_{esc})$ (black line). We observe a complex relation between event duration and accuracy, arising from two competing effects: while slower events tend to be more accurate in average behavior, longer escape times are often associated with polarity-loss events, leading to higher error rates.

For a larger noise ($\sigma = 0.5$, Fig 3B(ii)), we find that the speed–accuracy trade-off is almost flat compared to the low noise case. At the low $\beta_0$ limit, due to the large noise, cells lose polarity when migrating over the junction and have an almost equal probability of exiting along any of the three arms, leading to a $P(wrong) \sim 2/3$.

Taken together, these results indicate a trade-off between the speed and accuracy of cellular DDM during chemotaxis at a single junction. The origin of the speed–accuracy relation in our model, arises due to the slower cells having to spend longer time over the junction which allows the chemotactic bias to affect them for a longer period and therefore more significantly. In addition, slower cells have weaker intrinsic polarity, and are less persistent, so more easily respond to the external chemotactic bias. Fast cells are predicted to transiently form slow-mode shapes, extending long-lived protrusions along both the path toward the chemokine source and the "empty" path. This provides a possible explanation for the experimental observation that faster cells more often extend protrusions into both channels ("splitting"), even though the chemokine source is only along one of them [37]. We also find that this trade-off is sensitive to internal noise in actin treadmilling flows: at high noise levels, the effect is strongly diminished. In the next section, we examine how this behavior impacts large-scale chemotactic migration on a hexagonal network.

## Branched cell chemotaxis on a hexagonal network

We next study the migration dynamics of a branched cell on a hexagonal network in the presence of a global chemokine gradient imposed by a line source.

Our treatment of branched cells spanning multiple junctions follows [27] and is summarized in Theoretical model. Here, cells are exposed to a chemokine concentration field of exponential form,

$$c(r) = c_0 \, \exp\left(-\frac{r}{r_0}\right) \qquad (14)$$

where $c_0$ is the chemokine concentration at the source, $r_0$ is the decay length, and $r$ is the distance from an arm tip to the source. The enhancement of actin activity at each arm tip is given by

$$\beta(r) = \beta_0 \left[1 + \epsilon \left(1 + \frac{C}{c_0}\right) \frac{c(r)}{c(r) + C}\right] \qquad (15)$$

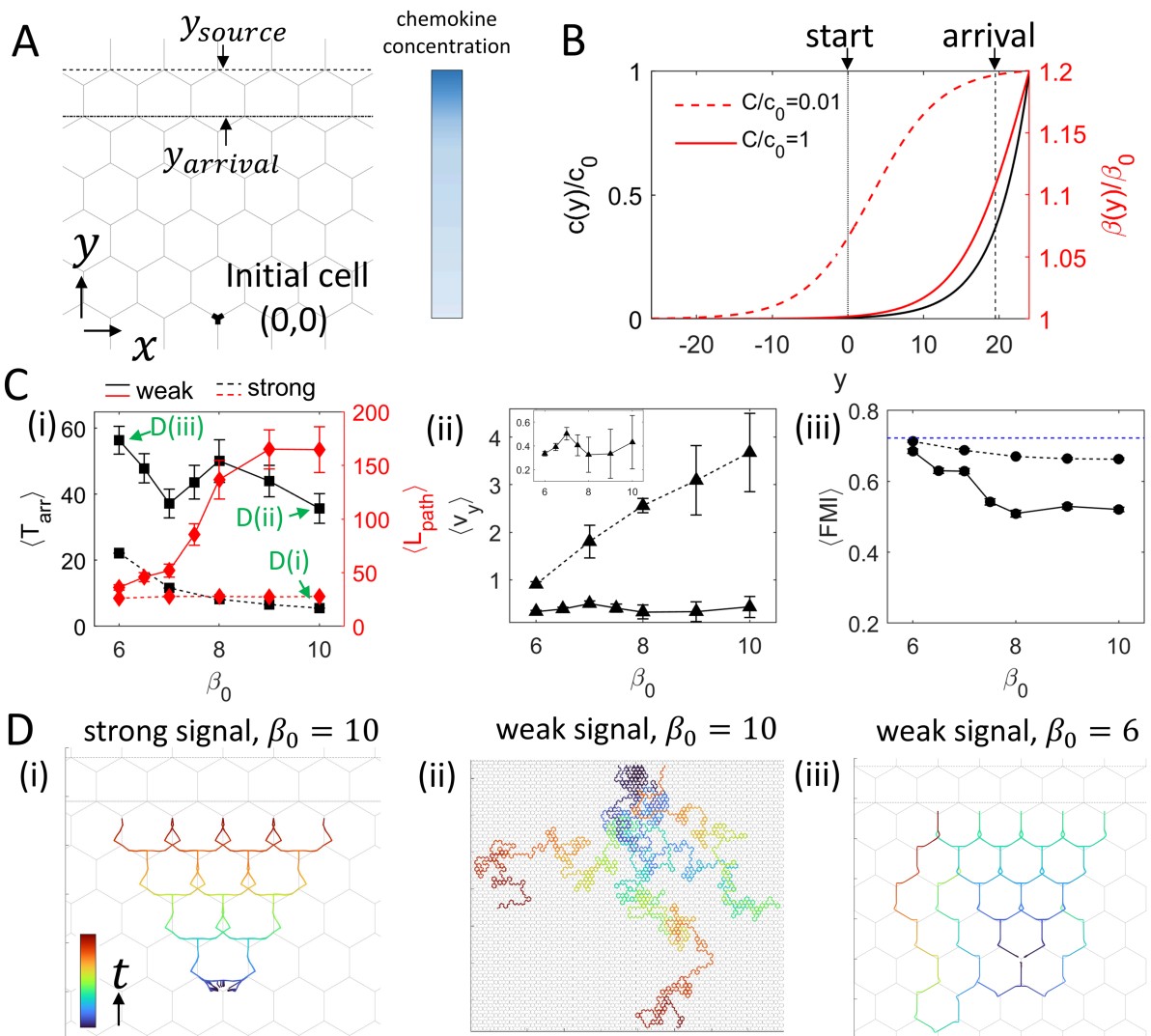

where $\beta_0$ is the baseline actin activity (no external cue), $\epsilon$ is the maximal relative enhancement, and $C$ is the saturation concentration. The prefactor $(1 + C/c_0)$ normalizes the enhancement to be maximal at $r = 0$.

Here we model a one-dimensional chemokine source located at $y_{source} = 8d$, where $d$ is the grid size (dashed line in Fig 4A). Arrival is defined as the first time any arm tip reached $y_{arr} = 6.5d$ (dash-dotted line in Fig 4A). The chemokine concentration profile is

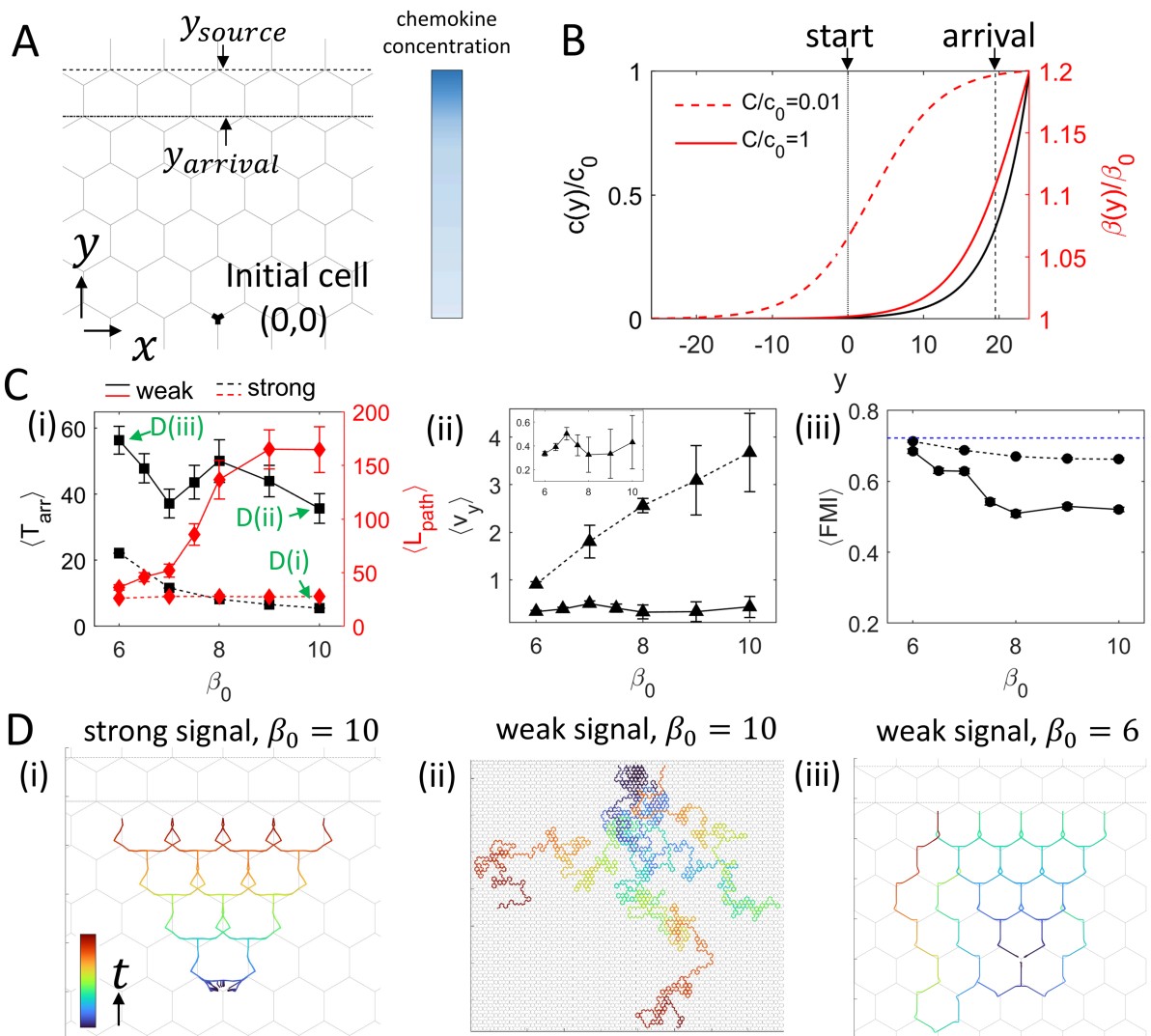

**Fig 4**. **Chemotaxis dynamics of cells on a hexagonal network with an exponentially decaying chemokine line source.** (A) Schematic of the model. The dashed line marks the chemokine source, and the dash-dotted line marks the arrival position. (B) Normalized chemokine concentration, $c(y)/c_0$ (black), and enhancement of actin activity, $\beta(y)/\beta_0$, versus $y$ in the strong-signal (red dashed) and weak-signal (red solid) regimes. Small $C/c_0$ (fast saturation) is effective at long range; large $C/c_0$ (slow saturation) is effective only near the source. (C) (i) Mean arrival time $\langle T_{arr}\rangle$ and path length $\langle L_{path}\rangle$ versus $\beta_0$. (ii) Mean C.O.M. velocity in the $y$-direction, $\langle v_y\rangle$, versus $\beta_0$. The inset corresponds to the weak-signal regime. (iii) Forward Migration Index, $\langle FMI\rangle$ (black solid), versus $\beta_0$. Black solid and dashed lines correspond to weak- and strong-signal regimes in (B), respectively. The blue dashed line denotes the theoretical maximum $FMI$ on the hexagonal network. We find a speed–accuracy relation (higher FMI for slower cells) for weak-signal (and low noise), which is greatly diminished for strong-signal. For the statistics, 100 and 500 stochastic simulations were performed for the strong- and weak-signal regimes, respectively. Error bars represent mean ± standard error. (D) Representative C.O.M. trajectories for (i) strong signal, $\beta_0 = 10.0$, (ii) weak signal, $\beta_0 = 10.0$, and (iii) weak signal, $\beta_0 = 6.0$. Maximal simulation time: $T_{max} = 1000$. Other key parameters: $\epsilon = 0.2$, $d = 3$, $\sigma = 0.1$.

$$c(y) = c_0 \, \exp\left(-\frac{|y - y_{\text{source}}|}{y_0}\right) \qquad (16)$$

where $y$ is the $y$-coordinate of the arm tip.

We fix the chemokine profile parameters ($\epsilon = 0.2$, $y_0 = 1.5d$) and vary $C/c_0$ to set the saturation regime. Specifically, $C/c_0 = 0.01$ corresponds to fast saturation (strong signal, effective at long range), whereas $C/c_0 = 1$ corresponds to slow saturation (weak signal, effective only near the source) (Fig 4B). At the beginning of each simulation, the cell is symmetrically positioned at the origin $(x, y) = (0, 0)$, allowed to spread, polarize, and migrate. A run ends upon reaching the source or at $T_{\text{max}} = 1000$.

We evaluate the efficiency of chemotaxis using the mean arrival time $\langle T_{arr} \rangle$ (Fig 4C(i)), the mean path length of the C.O.M. trajectory $\langle L_{path} \rangle$ (Fig 4C(i)), the mean speed towards the chemokine source $\langle v_y \rangle$ (Fig 4C(ii)), and the mean forward migration index, $\langle FMI \rangle$ (Fig 4C(iii)). The C.O.M. trajectory is sampled every $\Delta t = 0.01$; $L_{path}$ is computed as the cumulative arc length of this trajectory; $\langle v_y \rangle$ as the mean $y$-displacement per $\Delta t$; and $FMI$ as the net displacement divided by $L_{path}$. For the small fraction of runs ($\sim 1\% - 2\%$) that did not arrive within $T_{\text{max}}$, we assign $T_{arr} = T_{\text{max}}$.

In the strong-signal regime, cells rarely make wrong turns and follow directed paths toward the source (see typical trajectories at $\beta_0 = 10$ in Fig 4D(i)). Consequently, $\langle L_{path} \rangle$ is small and nearly independent of $\beta_0$ (Fig 4C(i), red dashed), $\langle v_y \rangle$ increases almost linearly with $\beta_0$ (Fig 4C(ii)), and $\langle T_{arr} \rangle$ decreases monotonically with $\beta_0$ (Fig 4C(i), black dashed), reflecting faster motion without loss of accuracy.

In the weak-signal regime, we find a more complex behavior as a function of $\beta_0$. At the lowest $\beta_0$ values, $\langle L_{path} \rangle$ and $\langle FMI \rangle$ are very similar to the strong-signal case (Fig 4C(i,iii)), indicating that these slowest cells can still maintain highly accurate, directed trajectories even under a weak signal. As $\beta_0$ increases, $\langle L_{path} \rangle$ grows monotonically (Fig 4C(i), red solid), indicating a higher propensity for wrong turns (away from the chemokine source). This trend is consistent with the speed–accuracy trade-off observed at a single junction (Fig 3B(i)), although the effect is less pronounced at the same noise level ($\sigma = 0.1$). The reduced migration accuracy at high $\beta_0 = 10$ is evident in the representative C.O.M. trajectories (Fig 4D(ii)). Note that a speed–accuracy trade-off was recently predicted using a different theoretical model of chemotaxis of branched cells [25].

At intermediate values around $\beta_0 \sim 7$, $\langle T_{arr} \rangle$ exhibits a pronounced minimum (Fig 4C(i), black solid), where gains in speed (Fig 4C(ii), inset) still outweigh the deterioration in directionality (Fig 4C(iii), solid). At higher $\beta_0$, errors rise faster than speed, yielding a maximum of $\langle T_{arr} \rangle$ at $\beta_0 \sim 8$ (Fig 4C(i), black solid). For even larger $\beta_0$, high speeds again dominate despite frequent wrong turns, lowering $\langle T_{arr} \rangle$ (Fig 4C(i,ii), black solid line).

In S3 Appendix, we examine how internal cellular noise ($\sigma$) and grid size ($d$) affect chemotactic migration. We find that increasing $\sigma$ impairs chemotaxis in both the strong- and weak-signal regimes. In addition, on relatively large grids, high-$\beta_0$ cells occasionally undergo slow-mode events along their trajectories. Note that a similar change in migration dynamics can also be obtained by keeping $d$ fixed while varying the cell length, for example, by adjusting the cell elasticity parameter $k$. We also find that the slow-mode events on large grids differ from those observed at single junctions (Fig 3) or on small grids (Fig 7). Instead of forming elongated arms in the migration directions, the cell undergoes a stick–slip event, after which one of the long arms extends backward, i.e., toward the direction from which the cell entered (S3 Fig).

Additional analysis linking the distribution of arrival times to wrong turns along the migration trajectories is provided in S4 Appendix.

From the theoretical results above, we summarize the following conclusions regarding strategies an immune cell may adopt to reach a wound or infection site via chemotaxis:

- In the presence of a strong signal, as may be expected close to the target site, cells with higher actin activity $\beta_0$ arrive faster (Fig 4C(i)), and therefore can prevent bacterial entry most effectively.

- In the presence of a weak signal, far from the target site, we find that the slowest cells exhibit relatively accurate and directed chemotaxis migration paths (Fig 4C(i,iii)). Despite the high accuracy, the arrival time for these slow-moving cells is relatively long (Fig 4C(i)).

- For weak signals, the fastest arrival times are found for a narrow range of intermediate actin activities ($\beta_0 \sim 7$) (Fig 4C(i)). In this intermediate regime, the chemotaxis is most efficient, as the cells maintain high accuracy while achieving sufficiently high speed for short arrival times (Fig 4C(i-iii)).

- For weak signals, high-$\beta_0$ cells meander more (lower chemotactic accuracy) yet still have a relatively short $\langle T_{arr} \rangle$ because of their high speeds (Fig 4C(i,iii)). On larger grids, transient slow-mode trapping can further delay arrival (S3 Appendix, S3 Fig), suggesting that efficient long-range recruitment of highly active cells may require mechanisms that enhance guidance cues.

- In general, we find that the speed–accuracy trade-off at single junctions (Fig 3B) is only weakly preserved for branched cell migration over the network, even under low-noise and weak-signal conditions. It is further diminished at high internal noise levels and in the strong signal regime (S3 Appendix).

## Comparison of the model with the chemotaxis of neutrophils

Next, we compare our theoretical model with *in vivo* data of neutrophils migrating toward laser-inflicted wounds (LW), including sterile wounds [5] and wounds infected with *Pseudomonas aeruginosa* in the fin of zebrafish larvae [29].

To simplify the analysis within our theoretical framework, we assume a spatially linear chemokine concentration profile along the *y*-axis. This corresponds to the strong-signal regime, since a linear profile provides a substantial guidance cue throughout the region of interest (S5 Appendix).

In Fig 5A we show an example of a neutrophil that initially moves away from the LW, which is located in the +*y*-direction. At $t = 0$ the LW is applied, after which the neutrophil responds to the chemical cue by transiently slowing down, pausing, and reorienting toward the LW. The time series of the cell's *y*-directed speed and its migration angle relative to the *y*-axis are shown in S6 Appendix. Once oriented toward the LW, the cell exhibits high speeds together with substantial directional variability. In Fig 5B we show the corrresponding simulation. At $t = 0$ the cell is migrating roughly away from +*y*, and a chemokine source positioned along +*y* is introduced. The simulated cell exhibits the same qualitative sequence: transient deceleration, reorientation and accelerated motion toward +*y*. During this process, our model describes how the cell effectively rotates its internal polarization and cytoskeletal protrusive activity, resulting in its reorientation to move towards the chemokine source (S6 Appendix). The role of such rotations in the cellular polarity during chemotaxis was recently shown in [42]. Note that we use a relatively high noise level $\sigma = 0.5$ in the simulations, which was previously found to be realistic when comparing the model to motile cells [26].

In Fig 5C, 5D we compare averaged dynamics from experiments and simulations (21 and 18 cells for panels C and D in [5], and 60 simulation runs). In Fig 5C, the average cell speed in experiments and simulations are plotted with respect to the LW time. In Fig 5D, we plot the average cell speed in experiments and simulations with respect to the time when the cell first began to move toward the source (the "beginning of movement", BM; see [5] and S4 Fig). To average out the effects of the hexagonal lattice geometry which induces large fluctuations in the velocity component orthogonal to the chemokine direction, we perform simulations on hexagonal grids rotated in 10° increments from 0° to 110° relative to the chemokine gradient (by symmetry, 120° returns to the original orientation), with five simulations at each angle. In the experiments, immune cells migrate through heterogeneous environments, so the velocity component orthogonal to the wound direction similarly averages out. The transient deceleration after the LW time and the acceleration after the BM time are observed in both experiments and simulations (Fig 5C, 5D). Overall, the simulations reproduce the qualitative features of the neutrophil response and provide a mechanistic explanation for these dynamics.

We next map the length and time scales between experiments and simulations. In the experiments, cells are approximately 20 μm long (Fig 5A) and move at 6 μm min$^{-1}$ on average (Fig 5C). Thus, the experimental cell covers about

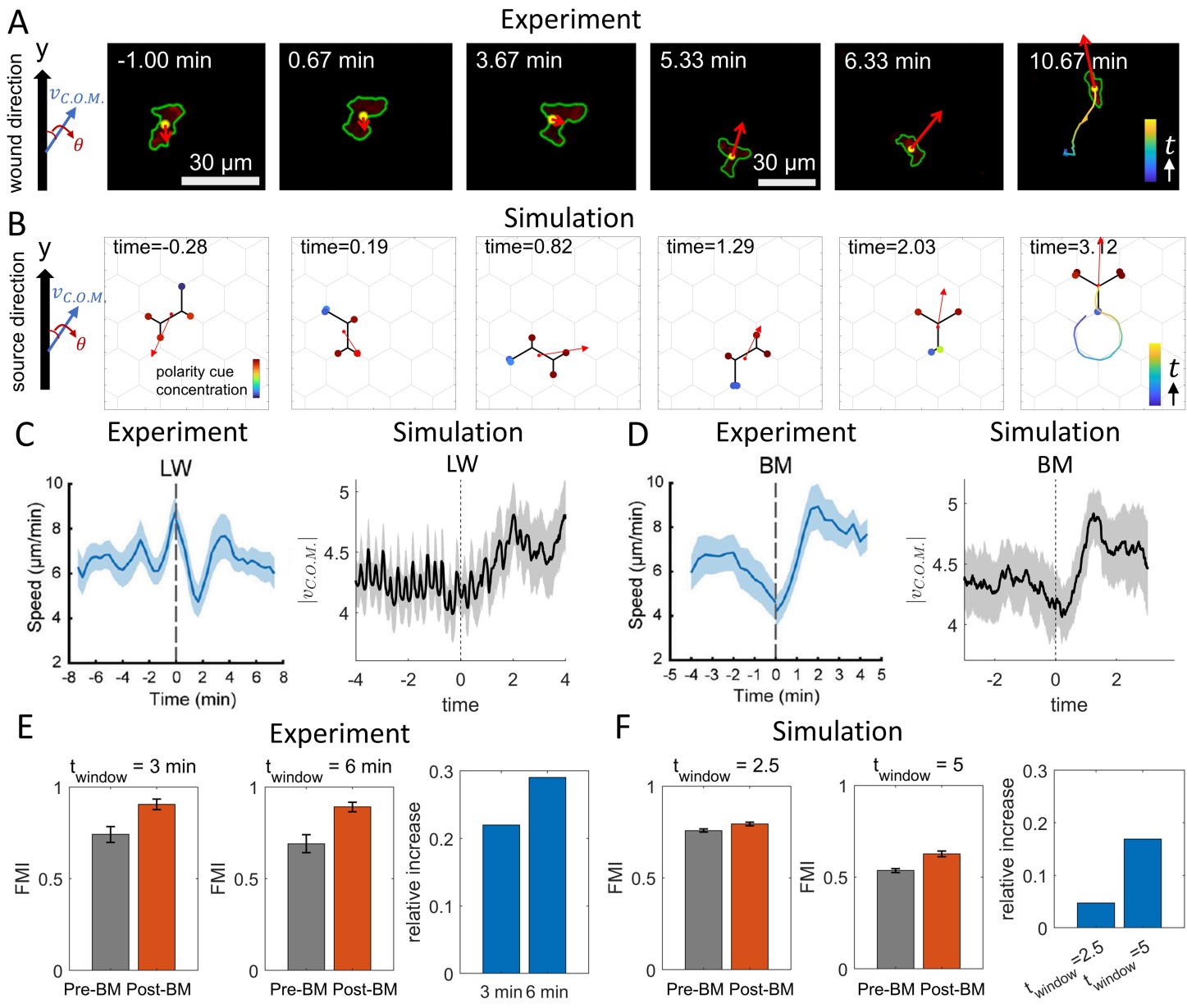

**Fig 5. Comparison of the chemotaxis dynamics between simulations and neutrophils in zebrafish larvae [5].** (A–B) Snapshots of experimental and simulated cell migration toward the chemokine source. Examples were selected where the cell migrated roughly away from the signal (located in the +y-direction; large arrow in the leftmost panels). At $t = 0$, a laser-inflicted wound (LW) was applied in the experiments, while a chemokine gradient was introduced in the simulations. The trajectory of the cell's C.O.M. is shown in the final panel, with red arrows denoting instantaneous velocity vectors. (C) Left: Experimental C.O.M. speed before and after the LW event (21 cells), adapted from [5]. Right: Simulated C.O.M. speed before and after the LW event (60 runs). (D) Left: Experimental C.O.M. speed before and after the beginning of movement (BM) (18 cells), adapted from [5]. Right: Simulated C.O.M. speed before and after BM (60 runs). Simulation data are obtained on hexagonal grids rotated by 10° in the range of 0° — 110° with respect to the chemokine gradient direction, with five runs performed for each angle. (E–F) Forward migration index (FMI) of the experimental and simulated C.O.M. within a defined time window (left two panels), and the relative increase after BM compared to before BM (right panel). Key parameters: $\epsilon = 0.1, d = 3, \beta_0 = 12, \sigma = 0.5$.

one cell length in $\sim 3$ minutes. The simulated cell ($\beta_0 = 12$) has average length 9.38 and speed 3.59 in simulation units and we correspondingly choose a simulation interval of $\sim 2.5$, over which the simulated cell also traverses about one

cell length. To facilitate the model–experiment comparisons, all data are presented as dimensionless or normalized quantities.

The effect of chemokine presence on the *FMI* of neutrophils for the two time windows used in [5] is shown in Fig 5E. For wild-type (WT) cells, *FMI* increases significantly after the BM time (by $\sim 20\% - 30\%$). Using the same model parameters $\epsilon, d, \sigma$ as in Fig 5, we plot *FMI* as a function of $\beta_0$ in S5 Fig. A substantial improvement in *FMI* due to the external gradient arises only at large $\beta_0 \sim 12$, corresponding to the regime that reproduces the WT observations. We therefore use this to calibrate $\beta_0$, and report in Fig 5F the change in *FMI* before and after BM in simulations.

We then compare the model predictions with the chemotaxis of cells treated with various drugs that inhibit the cytoskeletal activity. We incorporate inhibition of actin polymerization using CK666 by reducing the actin activity parameter $\beta_0$ (fit: $\sim 17\%$ decrease, consistent with the experimentally reduced cell length), and inhibition of myosin-II activity using blebbistatin by decreasing the contractile-stiffness parameter $k$ together with $\beta_0$ to reproduce the observed near-WT cell length (S7 Appendix). Comparisons with experiments on drug-treated cells (S7 Appendix) show good qualitative agreement when the WT cells are chosen to be in the high-$\beta_0$ regime of the model.

To clarify the relation between the scale and regimes of $\beta_0$ in our model and measured cellular behavior, we provide a qualitative mapping between $\beta_0$ and typical migration speeds (Fig 6), using the same calibration as in our previous work for human umbilical vein endothelial cells (HUVECs) and glioma cells [26,31]. In this framework, neutrophils are found to be located at higher $\beta_0$ than glioma and HUVEC, consistent with the experimentally observed hierarchy of migration speeds (Fig 6).

| Low activity regime | Medium activity regime | High activity regime |
|---|---|---|
| $\beta_0 = 0$        $\beta_0 \sim 5$ | $\beta_0 \sim 9$ | $\beta_0 = 15$ |
| - Cells get stuck on junctions. | - Cells move through junctions.<br>- Cells have lower persistence when highly branched.<br>- Cells move faster on the network and through junctions as $\beta$ increases.<br>- For the low noise, cells have lower directionality during chemotaxis with increasing $\beta$, but this can be masked by the noise. | - Cells move fast, but can get transiently trapped by slow-mode events.<br>- These slow-mode events can improve the directionality of chemotaxis. |
| **Examples of cells** | HUVEC<br>$\beta_0 \sim 6\text{-}7$, speed$\sim 1\text{-}2$ µm/min | Glioma<br>$\beta_0 \sim 8\text{-}10$, speed$\sim 3\text{-}4$ µm/min | Neutrophils<br>$\beta_0 \sim 10\text{-}13$, speed$\sim 10\text{-}20$ µm/min |

**Fig 6**. **Schematic mapping between the model's actin activity parameter $\beta_0$ and qualitative migration phenotypes.** Based on fitting cellular migration patterns with our model to observations of several cell types (HUVEC and glioma [26,31]), together with the fits shown in Fig 5 for neutrophils, we establish a mapping between the actin activity parameter $\beta_0$ and the observed migration speeds of different cell types. At the top we describe the main migration phases of branched cells as function of $\beta_0$ [26,27]: At low $\beta_0$ cells get trapped at junctions, intermediate $\beta_0$ enables robust migration with increasing speed as $\beta_0$ increases, and high $\beta_0$ produces fast motion with intermittent slow-mode events that can enhance directionality. At the bottom we give examples of cell types, with the range of values of $\beta_0$ that were fitted to them, and their range of observed persistent migration speeds. This figure illustrates a qualitative correspondence between the actin activity parameter of our model ($\beta_0$) that is consistent with the observed hierarchy of cellular speeds.

PLOS Computational Biology

Our conclusion that neutrophils correspond to a higher-$\beta_0$ regime than the other cell types considered here is further supported by their migration when starting farther from the LW (*In vivo* experimental methods). In this case, cells frequently exhibit highly meandering trajectories (Fig 7A(i)). Such meandering paths are more frequently observed in the high-$\beta_0$ regime in our model (Fig 4D). In addition, slow-mode events are often identifiable along these paths (Fig 7A(ii,iii)), consistent with the slow mode observed in the model at high $\beta_0$ (Fig 3E). The prevalence of these slow mode events is quantified in S6 Fig.

Very similar migration and shape dynamics features are observed for PLB-985 (promyelocytic leukemia blasts) cells migrating *in vitro* toward a linear chemokine gradient (Fig 7B), while confined within a regular hexagonal lattice of rectangular pillars (*In vivo* experimental methods). This suggests that the observed characteristics do not rely on specific *in vivo* interactions of neutrophils with surrounding tissues, but rather reflect general features of fast-moving branched cells in complex environments. Simulations of high-activity cells migrating toward the LW also predict meandering paths with slow-mode events (Fig 7C). In both experiments and simulations, the total cell length increases during slow-mode

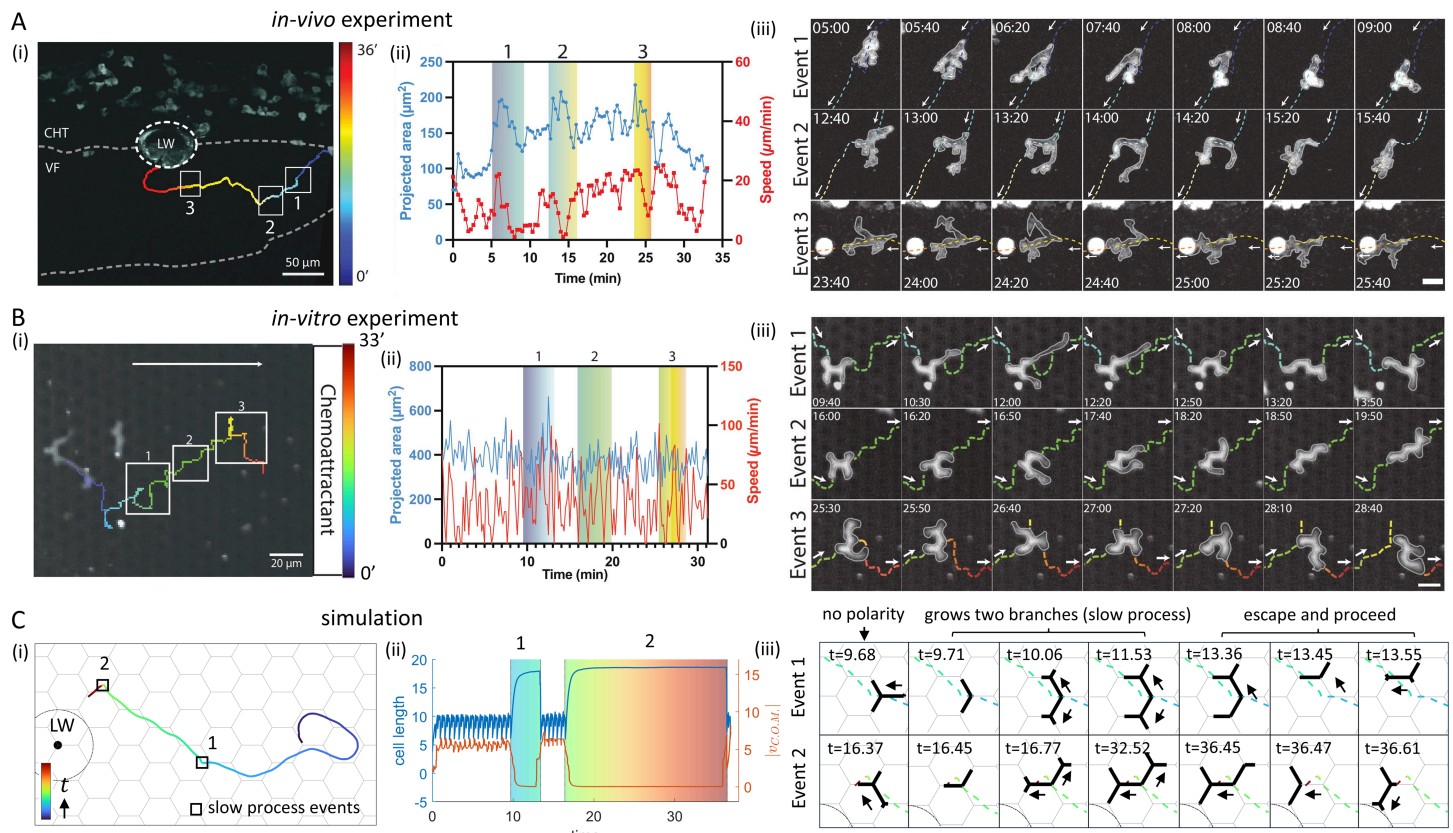

**Fig 7**. **Comparisons of the chemotaxis characteristics between simulations and neutrophils far away from the LW.** (A) (i) Representative trajectory of a neutrophil migrating toward the LW, with trajectory color denoting migration time. White boxes highlight time intervals along the trajectory where slow mode events occur, corresponding to the colored regions in (ii) and the snapshots in (iii). (ii) Dynamics of the projected area and overall speed of the neutrophil during migration. (iii) Snapshots of the neutrophil during the three slow mode events. (B) Same panels as in (A), for a PLB-985 (promyelocytic leukemia blast) cell migrating *in vitro* up a linear chemokine gradient while confined within a regular pillar lattice. (C) (i) Representative simulation trajectory of a cell migrating toward the LW, with trajectory color denoting migration time. Black boxes highlight time intervals along the trajectory where slow mode events occur, corresponding to the colored regions in (ii) and the snapshots in (iii). (ii) Dynamics of cell length and C.O.M. speed during migration. (iii) Simulation snapshots of the cell during the slow mode events. Parameters: $C/c_0 = 0.01, \epsilon = 0.1, d = 3.7, \beta_0 = 12.6, \sigma = 0.6$.

episodes as two arms extend, before one retracts and migration resumes (Fig 7A–7C(ii)). In the simulations, slow-mode events tend to be more persistent closer to the source, due to the increased actin activity (Fig 7C).

## Discussion

We present here a theoretical model for chemotaxis of branched cells on networks. The model exhibits a speed–accuracy trade-off for cell chemotaxis on a single junction in the limit of very low internal noise in actin polymerization activity (the $\sigma$ parameter; Fig 3B(i))). At higher (more realistic) noise levels, this trend becomes weak (Fig 3B(ii)). Extending to network migration, we observe at most a mild effect in the low-noise regime, where slower cells have straighter paths (higher FMI, Fig 4C(iii)). However, this is again largely lost as noise increases (S2 and S5 Figs). Overall, the speed–accuracy trade-off is primarily a low-noise feature of the model and is further reduced in the network setting. A somewhat similar speed–accuracy tradeoff was recently observed in another theoretical model [25], where cells with a larger number of protrusions are slower and more accurate, in qualitative agreement with our model (where more highly branched cells are slower [27]). Future experiments may investigate the theoretically-predicted speed–accuracy tradeoff.

The model provides a mechanistic description of how motile cells respond, reorient and migrate towards a chemoattractant source, in good agreement with experimental observations of immune cells (neutrophils) *in vivo*. Note that our present model does not describe the important role of microtubules during cellular branching dynamics and directional decision-making [18,20,21,43], and this feature will be added in the future.

Comparing the model to experimental observations of neutrophils migrating *in vivo* to the site of a laser-inflicted wound in a zebrafish larva fin, and to PLB-985 cells migrating *in vitro* within a regular lattice of pillars, suggests that these immune cells correspond to the fast-cell limit of the theoretical model (Fig 6). Our model predicts that cells in this limit have the advantage of minimizing their arrival time to the target site, if they start close to the wound and receive a strong chemokine signal. This can therefore be a good strategy if the neutrophils are uniformly spread at sufficiently high density in the tissue, so that there are always some cells close to any wound or infection site. However, our model predicts that such fast moving cells perform less accurate chemotaxis when further from the wound, where the original wound-secreted signal is weak. For efficient recruitment of these far-field neutrophils the immune cells need to utilize different mechanisms, such as secretion of their own chemokine signals. Indeed, neutrophils have evolved such recruitment mechanisms, which have been discovered and are being studied [44–47]. These results of our model regarding neutrophils fit very well with their functional role of "first-responders" that engage with bacteria invasion at wound sites [29]. There may be scenarios where cells would rather maximize accuracy at the price of moving more slowly. This may be relevant, for example, for immune cells that need to reach a stationary target such as a tumor, or for cells performing chemotaxis during development.

We demonstrate here that our simplified theoretical framework provides a description of the underlying mechanisms that drive the changes to the cell polarization (rotation and reorientation [42]) and shape changes in response to a chemotactic signal. The same model also allows us to explain the large-scale migration patterns, of highly branched cells during chemotaxis in complex geometries. This can be a useful tool for deciphering the behavior of motile cells, such as immune or cancer cells, performing chemotaxis inside the complex geometries of living tissues. As we show above, the model allows us to connect observations of single cell shape and polarity dynamics to the large-scale chemotaxis behavior.

## Supporting information

**S1 Fig. Microfabricated PDMS devices design.** (A) Blue dots indicate loading areas for cells and chemoattractant. Loading areas are surrounded by an area of circular pillars, which serve as an antechamber from where cells enter the 0.7x2.5 mm analysis area. (B) Analysis area harbors 10x10 $\mu m$ rectangular pillars of 3.8 $\mu m$ height. Pillars are distanced 5 $\mu m$ and 3 $\mu m$ in vertical vs. horizontal direction, and arranged in a hexagonal lattice.
(TIFF)

**S2 Fig. Effect of the slow mode on decision-making accuracy for the system in Fig 3B(i).** (A) Mean probability of entering the slow mode at the junction, $P(slow)$ (black line), and the error rates of the fast and slow processes (red and green lines, respectively). (B) (i–iii) Distributions of escape time $T_{esc}$ (histograms) and error probability $P(wrong)$ (black line) as functions of $T_{esc}$, for the $\beta_0$ values indicated by the vertical dashed lines in (A). Key parameters: $\epsilon = 0.001$, $\sigma = 0.1$. (TIFF)

**S3 Fig. The second type slow mode during cell migration on large grids.** (A) Representative simulation trajectory of a cell migrating toward the LW, with trajectory color denoting migration time. Black boxes highlight time intervals along the trajectory where slow mode events occur, corresponding to the colored regions in (B) and the snapshots in (C). (B) Dynamics of cell length and C.O.M. speed during migration. The three colored regions correspond to the slow mode events marked in (A). (C) Simulation snapshots of the cell during the slow mode events. Parameters: $C/c_0 = 0.01$, $\epsilon = 0.2$, $d = 9$, $\beta_0 = 12$, $\sigma = 1.2$. (TIFF)

**S4 Fig. The "beginning of movement" (BM) time.** Time evolution of $v_{y,C.O.M.}$ for six randomly selected simulation trajectories. Black dots mark the BM time for each trajectory, i.e., the moment the cell first begins to move toward the source (positive $y$-direction). (TIFF)

**S5 Fig. *FMI* as functions of $\beta_0$.** Gray and red correspond to the *FMI* before and after the BM time, respectively. Key parameters: $\epsilon = 0.1$, $d = 3$, $\sigma = 0.5$. (TIFF)

**S6 Fig. Characterization of "slow mode" events for cells *in vivo* (as in Fig 7A).** GCamp6F transgenic zebrafish embryos (3 dpf) were infected with *Pseudomonas aeruginosa*, laser-wounded, and imaged using a two-photon confocal microscope, as previously described. Neutrophil trajectories were manually tracked using the IMARIS software. Trajectories were classified as meandering or straight based on the *FMI* quantified by the software (>0.60 considered straight). They were further categorized by the presence or absence of slow mode events. The graphs show both the absolute counts (left) and percentages (right) of cells displaying slow mode events within the straight and meandering categories. Percentages are based on 39 and 11 cells, respectively, as indicated. Videos with overall recruitment of more than 15 cells were selected. In total, $N = 46$ neutrophil trajectories from 7 independent videos/embryos were analyzed. $****$ $P < 0.0001$, Chi-square test (and Fisher's exact test). (TIFF)

**S1 Appendix. Calculation of the steady-state distribution of the polarity cue concentration and the local actin treadmilling flows.** (PDF)

**S2 Appendix. Onset of the stick-slip motion for cells on a one-dimensional line.** (PDF)

**S3 Appendix. Effect of cellular internal noise and grid size on chemotaxis dynamics.** (PDF)

**S4 Appendix. Detailed analysis of cell trajectory dynamics in the weak-signal regime.** (PDF)

**S5 Appendix. Comparing the experimental response of the cell to the chemokine with the model.** (PDF)

**S6 Appendix. Analysis of the cell's response to the chemokine gradient in experiments and simulations.**
(PDF)

**S7 Appendix. Comparison of the chemotaxis characteristics of drug-treated cells between simulations and neutrophil experiments.**
(PDF)

**S1 Code. Fortran 90 code package (ZIP) containing two programs: (i) simulations of cellular decision-making at a single junction, and (ii) branched-cell migration on hexagonal networks.**
(ZIP)

## Author contributions

**Conceptualization:** Nir S. Gov.

**Formal analysis:** Jiayi Liu, Jonathan E. Ron, Giulia Rinaldi, Ivanna Williantarra, Antonios Georgantzoglou, Ingrid de Vries.

**Funding acquisition:** Michael Sixt, Milka Sarris, Nir S. Gov.

**Investigation:** Giulia Rinaldi, Ivanna Williantarra, Antonios Georgantzoglou, Ingrid de Vries.

**Methodology:** Jiayi Liu, Jonathan E. Ron, Nir S. Gov.

**Project administration:** Nir S. Gov.

**Resources:** Michael Sixt, Milka Sarris.

**Software:** Jiayi Liu.

**Supervision:** Michael Sixt, Milka Sarris, Nir S. Gov.

**Visualization:** Jiayi Liu, Antonios Georgantzoglou.

**Writing – original draft:** Jiayi Liu, Nir S. Gov.

**Writing – review & editing:** Jiayi Liu, Jonathan E. Ron, Giulia Rinaldi, Ivanna Williantarra, Antonios Georgantzoglou, Ingrid de Vries, Michael Sixt, Milka Sarris, Nir S. Gov.

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
