## [Decision Letter · Decision Letter 0]

17 Sep 2025

PCOMPBIOL-D-25-01385

Modelling chemotaxis of branched cells in complex environments provides insights into immune cell navigation

PLOS Computational Biology

Dear Dr. Liu,

Thank you very much for submitting your manuscript to PLOS Computational Biology. After careful consideration, we feel that it has merit but does not fully meet PLOS Computational Biology's publication criteria as it currently stands. Therefore, we invite you to submit a revised version of the manuscript that addresses the points raised during the review process.

Please submit your revised manuscript within 60 days Nov 17 2025 11:59PM. If you will need more time than this to complete your revisions, please reply to this message or contact the journal office at ploscompbiol@plos.org. Please include the following items when submitting your revised manuscript:

We look forward to receiving your revised manuscript.

Kind regards,

Calina Copos, Ph.D.

Academic Editor

PLOS Computational Biology

Dimitrios Vavylonis

Section Editor

PLOS Computational Biology

**Journal Requirements:**

At this stage, the following Authors/Authors require contributions: Jiayi Liu, Jonathan E. Ron, Giulia Rinaldi, Ivanna Williantarra, Antonios Georgantzoglou, Ingrid de Vries, Michael Sixt, Milka Sarris, and Nir S. Gov. Please ensure that the full contributions of each author are acknowledged in the "Add/Edit/Remove Authors" section of our submission form.

3) We noticed that you used the phrase 'unpublished data' in the manuscript. We do not allow these references, as the PLOS data access policy requires that all data be either published with the manuscript or made available in a publicly accessible database. Please amend the supplementary material to include the referenced data or remove the references.

4) We do not publish any copyright or trademark symbols that usually accompany proprietary names, eg ©,  ®, or TM  (e.g. next to drug or reagent names). Therefore please remove all instances of trademark/copyright symbols throughout the text, including:

- © on page: 4.

5) We notice that your supplementary figures are uploaded with the file type 'Figure'. Please amend the file type to 'Supporting Information'. Please ensure that each Supporting Information file has a legend listed in the manuscript after the references list.

6) We notice that your supplementary Table, and information are included in the manuscript file. Please remove them and upload them with the file type 'Supporting Information'. Please ensure that each Supporting Information file has a legend listed in the manuscript after the references list.

7) When completing the data availability statement of the submission form, you indicated that you will make your data available on acceptance. We strongly recommend all authors decide on a data sharing plan before acceptance, as the process can be lengthy and hold up publication timelines. Please note that, though access restrictions are acceptable now, your entire data will need to be made freely accessible if your manuscript is accepted for publication. This policy applies to all data except where public deposition would breach compliance with the protocol approved by your research ethics board.

8) In the online submission form, you indicated that "The code that support the findings of this study are available from the authors upon reasonable request."  All PLOS journals now require all data underlying the findings described in their manuscript to be freely available to other researchers, either

1. In a public repository

2. Within the manuscript itself

3. Uploaded as supplementary information.

9) Please include the grant numbers in the Funding Information tab ensuring that the funders and grant numbers match between the Financial Disclosure field and the Funding Information tab in your submission form. Note that the funders must be provided in the same order in both places as well.

10) We see that your study includes live participants. Please ensure that your ethics statement is included under a subheading 'Ethics Statement', at the beginning of your Methods section.

Note:  The Ethics Statement should include : The full name(s) of the Institutional Review Board(s) or Ethics Committee(s), and the approval number(s), or a statement that approval was granted by the named board(s).

**Reviewers' comments:**

Reviewer's Responses to Questions

Reviewer #1: The authors investigate the motion of neutrophils in a "complex" environment (in vivo in zebrafish, in vitro with pillars, and simulated in hexagonal arrays of micro-patterns). They compare experimental data to a model of the "branching cells" that is basically a set of spring-like arms competing for dominance. The eventual takeaway is that there is a a speed-accuracy trade-off.

The authors show simulations for a single cell at a junction point and later for a single cell in a hexagonal array. They compare various model predictions to the behaviour of real neutrophils in the experiments.

Major issues

Overall, while the title and abstract were very enticing, the paper as a whole was, at its current state, less interesting than expected. Perhaps the authors assumed that everyone is familiar with their (previously published) model, and so spent little attention on motivating the assumptions and assessing both benefits and shortcomings. Perhaps the take-away speed-accuracy trade-off seems unsurprising. Or perhaps, the overall presentation could have used more thoughtful self-critical analysis and better discussion, to avoid a reader's reaction of "so what?". It is overall not a reader-friendly presentation. Probably, the authors could improve the writing to make this into the interesting paper that it deserves to be.

The description of the model in the main text is cursory, and anticipates that the reader will be familiar with refs [18, 26]. The main assumptions should be here (not only in the SI) and justified as key properties that are necessary and sufficient for describing neutrophils.

Review of work on cells in mazes, cells creating or adapting the local gradients, and other models for cell arms based on competing springs was not very complete.

The authors' discussion of tests of the model are summarized in a quick statement on L413-415, and in Fig S7.

"We incorporate the inhibition of actin polymerization or myosin-II activity in our model through changes to the model parameters, as was done previously [18]."

The values, of parameters, but not a justification, is given in Fig S7. The assumed effects of the various inhibitors is not explained. Again, the paper should be self-contained.

In general, there are a great many graphs and bar charts, but the overall message is hard to parse. The figures are hard to digest because they have series that go horizontally as well as vertically in the same figure, the caption are very terse, and there is a lot of notation to memorize. The abbreviations and notation make it hard to get the message of what the authors intend to convey. A short conclusion or significance statement in the captions could be helpful.

Figure S1 seems to be missing?

The Discussion lacks caveats and comparisons of the model with others in the literature. For example, how important is the specific set of assumptions about the friction and the dynamics of the adhesion bonds? The rest of the model is fairly simple linear kinetics, so it would be good to briefly comment on how important the form of equation (6) is.

Minor points

L 39 - ...branching is determined by the network geometry.

L 45 - ... a range of chemical gradient generated by tissue – cells or themselves.

L 54 - …indecision (not in-decision)

Cell polarization in the model is mentioned on L 195, but it is not explained what this means in terms of the model. What aspect defines that polarization? There is something related to this in Fig S2, but it is not referenced here and that figure is confusing.

If I understood it, the chemokine gradient is linear with the y coordinate of the location, which assumes that the hexagons are entirely permeable to the chemokine. But if the hexagons are impermeable, then the chemokine would be graded along the path lengths. I am wondering if this might help the cells chose the most direct (shortest) paths? Can the authors please comment?

Fig 2 shows that total cell length can vary between 3-13 units. Can the authors please comment on how much the real cells stretch in the experiment.

The "critical polarization length" L_c is mention on L 242 and 244 but does not seem to appear in the model (appendix S1)? The value itself seems to be dependent on beta as it appears to take different values in Fig 2 C(i) and C(ii).

"Energy" is mentioned once on L 358. It was not clear where this statement came from.

FMI (line 319) should be defined somewhere.

You use CK-666 and CK666. Please pick one. Searching for either one does not lead to any description of what this inhibitor is doing.

Reviewer #2: In this paper the authors applied a mathematical model (that describes cells extending branches to probe their environment) to chemotaxis of neutrophils. In the model, the chemokine enhances local actin activity and thus drives the directed motion. It was found that chemotacting cells exhibit a speed-accuracy trade off, where slow cells can more accurately respond to weak gradients. The model and the two experiments showed similar migration characteristics, indicating general features for neutrophils that are well-described by the model.

General comments

The authors did a very thorough analysis of the model. Furthermore, the model was compared to not one but two experimental set-ups. Showing that the model is well applicable.

Because of the wealth of quantification, figure panels and supporting text and figures, the main message sometimes got lost. I am not sure what to recommend to improve this. The introduction and discussion are relatively short. The impact of this paper could be improved by explaining the broader context and relevance some more. The explanation of the mathematical model can be significantly improved (see specific comments for appendix S1).

Specific comments

Abstract:

Incorrect english / typing mistake in second sentence of abstract “such chemotaxis…”

“of laser-inflicted wound” I think should be “of A laser-inflicted wound”

line 38: if determined → is determined

line 45: generated in by → generated by

Methods section:

I would personally prefer it if the explanation of the mathematical model and its equations are part of the methods and material section, instead of in the appendix S1. Figure S2 too.

Figure S2 mentions a polarity cue that is not mentioned in Appendix S1. It is as far as I can tell, only firstly mentioned in the main text in Figure S6. I am not sure if this is an additional variable in the model and how it is regulated. Is it c_i(x)?

Results section:

line 219-line 222: can you explain based on the model equations and assumptions why for this range of beta values, the cell starts to stick-slip?

Line 297-301: how are Lpath, average v_j and FMI calculated?

Figure 3C: the green arrows, should they indicate D(i) to D(iii) instead of E(i) to E(iii)?

Lie 372-375: Why did the chemokine concentration have to be linear now? And how does the chemokine profile relate to the in-vitro experiment chemical gradient (if known?)

Line 416: Can you briefly mention which parameters were changed and why?

Line 683: does this refer to the right Figure? I do not see a figure 5E and Supp Figure5E does not seem to relate (unsure).

Caption Figure 5 panel C(ii): The two colored regions correspond to the three slow mode

events marked in → The two colored regions correspond to the two slow mode

events marked in

Discussion section:

What are the model limitations, what can be improved? How do the model results compare to existing literature?

Appendix:

In S1 Appendix, some details are missing for me. The equations are there but the assumptions are not explained well. Without this, it’s hard to gain some intuition and relate it to the biology. I recommend to explain the model more step by step, each term and each parameter and the underlying assumptions. Examples if what is unclear:

What is k in equation 5? Should it be kappa? What role does the term (L-1) play, is L normalized to 1?

Line 572: is kappa the same as k?

I don’t understand what variable describes the direction of the motion of the arm’s leading edge, which apparently is part of equation 8 somehow? Why does the equation 6 and 8 look like this? Is it modeling slip-bond dynamics? What do the two terms in equation 6 describe? What is r in equation 6? What role does n_i play in equation 8? It’s not easy to understand the underlying physics or assumptions in particular in equation 8. Why is the Heaviside function used?

Equation 9: same kind of questions, why this form, explain the dynamics that is described in equation 7.

slip-bond dynamics: many bonds are now know to behave as catch-bonds too. Do these type of adhesive bonds exhibit catch-bond dynamics in real life and if so, how would it affect the model results?

Line 587 to Line 590: how is the calculation performed? Why is it spatially distributed in the model and not constant? Do these exponential sections mean that the concentration goes exponentially up or down towards the tip?

S3 and S4 Appendix have the same title. Probably a mistake?

Table S1: “some of the model parameters used in this study” are not all of them included in the table?

I would recommend to add two columns: a short description of the meaning of the parameter, or name of the parameter, and one with the unit of the parameter.

Reviewer #3: In this paper the authors formulate a model for chemotaxis of cells having a branched geometry wherein the authors have extended some of their previous work on motility of branched cells to account for bias in cell motility due to chemotaxis. The primary claim of this paper, which has been shown clearly from the mathematical model is that slow-moving cells can chemotax better (Fig 2B, Fig 3C(iii)). This is an extension from their Nature Physics article of 2024, and the modelling is well done. The authors compare the model with chemotaxis of immune cells both in-vitro and in-vivo. This is just what would justify a PLOS computational biology paper. The data are fairly clear.

However, it is not clear how the experimental data shown in this paper are in support of the speed-accuracy trade-off hypothesis that is predicted by the model. In fact the only discussion of this hypothesis is provided in the supplementary figure 7 of the paper. Given that this is one of the key novelties of the paper, I would have expected this to be a key figure of the main text and a decent discussion of the same in the main text. However, the only discussion that the authors provide for this is in the form of a single sentence in line 417-418 where the authors write “Comparing the model to the 416 experiments on these drug-treated cells (S7 Fig) gives good qualitative agreement, and points to the WT cells residing in the high-β0 regime of our model”. The authors have not provided any justification for this claim of ‘qualitative agreement’, and as far as I can observe (see below), there isn’t any agreement as far as the speed-accuracy trade-off hypothesis is concerned.

In summary, the experimental data presented in this paper does not go far enough to validate verify or validate the key hypothesis presented by the model, meaning that the conclusions (clearly stated in the Abstract) are not yet supported by the data. I therefore cannot recommend publication of this manuscript in the current form in PLOS computational biology. If the authors can obtain data that better supports the model, or justify much more effectively why their data provides such support, the paper would be well suited. Alternative would be to address some of the below points regarding presentation of the model, and publish the model, which is well made and of interest, independent of the experiments in a more specific journal of mathematical biology.

Specific comments:

- In the experiments, the cells are treated with drugs which either inhibit actin polymerization or myosin-II activity, and the only conclusion that can currently be drawn from these experiments (in-line with previous experiments on cell motility), is that these drugs reduce the overall motility of the cells. None of the results presented in Figure S7 as presented show that the drug-treated cells can chemotax better than the normal ones, and this is something which is also in line with previous experiments. Maybe the authors disagree and can explain more effectively what the data supports the model. But the current experimental data do not support the novel conclusion.

- In lines 207-208, the authors refer to a ‘noise’, it is not clear where this noise comes from, and how it manifests itself in the cell motility. I would expect to see the mathematical equations or some intuitive explanation of this noise term before I see any results where it is varied.

- The authors refer to a ‘slow mode’, as something which occurs if the actin-activity parameter independent of the chemoattractant β_0 increases – this is something which they claim to have found in a previous paper but I struggle make any sense of Figure 2B(i) and I believe I am more expert than most readers. Could the authors offer an intuitive physical or mathematical explanation of the same.

- In Figure 2B(i), it could be more appropriate having discrete data points (something like 2C(ii)) because the data presented here is from individual simulations, and not an analytical formula. Furthermore, it is important to mention how many stochastic simulations have been considered while reporting a single data point, and also report the error associated with the data due to stochasticity.

- In Figure 2B(i), it seems that beyond β_0>11, the cells are unable to make a decision within the simulated time, which could make the definition of P(wrong) ambiguous. My recommendation would be to clarify the definition of these probabilities more carefully.

- In Figure 4, the authors compare the experimentally observed cell speeds with the y-component of the cell velocity obtained from simulations. I do not understand why it is justified to do that. What this means is that you are comparing one aspect of the experiment with a separate quantity from the model, which appear to be a meaningless comparison. The authors must explicitly justify why they do it or use a fair comparison.

small errors:

not defining parameters before discussing it (FMI for example),

assuming a reader would have read their previous papers to make sense of the notation that they used previously (eg the stick-slip difference).

Please focus on the presentation of the mathematical model to make it clearer and revise the presentation.

**Have the authors made all data and (if applicable) computational code underlying the findings in their manuscript fully available?**

Reviewer #1: **No:** the authors would make their code available "upon reasonable request"

Reviewer #2: Yes

Reviewer #3: None

PLOS authors have the option to publish the peer review history of their article (what does this mean?). If published, this will include your full peer review and any attached files.

Reviewer #1: No

Reviewer #2: **Yes:** Elisabeth Geraldine Rens

Reviewer #3: No

**Figure resubmission:**
---

## [Decision Letter · Decision Letter 1]

17 Dec 2025

PCOMPBIOL-D-25-01385R1

Modelling chemotaxis of branched cells in complex environments provides insights into immune cell navigation

PLOS Computational Biology

Dear Dr. Liu,

Thank you very much for submitting your manuscript to PLOS Computational Biology. After careful consideration, we feel that it has merit but a couple reviewers have asked for several points to be addressed prior to their recommendation for publication. Therefore, we invite you to submit a revised version of the manuscript that addresses the points raised during the review process.

We look forward to receiving your revised manuscript.

Kind regards,

Calina Copos, Ph.D.

Academic Editor

PLOS Computational Biology

Dimitrios Vavylonis

Section Editor

PLOS Computational Biology

**Journal Requirements:**

1) In the online submission form, you indicated that "Data are available from the corresponding author upon request." All PLOS journals now require all data underlying the findings described in their manuscript to be freely available to other researchers, either

1. In a public repository

2. Within the manuscript itself

3. Uploaded as supplementary information.

2) Please include the grant numbers in the Funding Information tab.

**Reviewers' comments:**

Reviewer's Responses to Questions

Reviewer #1: The authors have addressed most of the technical issues in the paper, so the presentation is improved. There are some typos, (including chamotaxis, chemotxis).

That said, my overall impression is that (1) The branching cells model was motivated by experiments (Fig 1, Fig 5A, Fig 6), though the model predictions are not closely compared to those experiments. (2) the model is somewhat arbitrary description of generic cell motility, so may have limited applicability beyond this system; if it agrees well with the experiments, this should have been more strongly demonstrated with various statistical analyses. (3) The authors draw some conclusion on speed vs accuracy tradeoff with noise, but, while there are a lot of specific detailed results on simulation runs, it is hard to understand what aspect of this model lead to those results.

For these reasons, I find it hard to make a recommendation for this paper.

Reviewer #2: The article has been improved. The authors have answered most, but not all, of my concerns.

In the revised article, the authors mention that the speed-accuracy trade-off is much less pronounced in the presence of realistic levels of noise. I certainly agree with the addition of this important caveat, as indicated by one of the other reviewers. But now it makes me wonder what is the added value of this paper if it’s main message (this trade-off) doesn’t actually occur for realistic parameter values.

I find the discussion section still very short. Here I would expect a discussion on the biological relevance of the speed-accuracy trade-off. What do we learn from this trade-off that was found in the model, even though cells may not actually enter this ball-park of low noise in real life? Or are there certain situations/conditions in which they do?

Perhaps you can relate it to what is mentioned in the introduction in line 62 to line 68 and in line 540 to line 548 in the discussion section.

Line 62-68: I noticed now that this is a very long sentence, please break it up in more sentences. Currently, is also reads as if you discovered in your simulations what are these constraints for efficient neutrophil response “they must be uniformly distributed at high density throughout the tissue, while having mechanisms for enhancing the chemical signals that facilitate efficient recruitment of neutrophils from larger distances.” but it’s more of a hypothesis that extended from your result, right? And not a direct result from your simulations?

Reading about the trade-off existing only in low noise conditions, also made me wonder when would a slow but accurate path/trajectory be beneficial? If cells just need to arrive at, say, a wound, would it matter how much they meander before they arrive, or is only a fast arrival time necessary? If, say, a cell is chasing a bacteria, perhaps less meandering and more accuracy would be more favorable? Could cell migration noise somehow be targeted by some drug to get cells in this high accuracy and low speed regime, or in the high speed low accuracy regime? Perhaps commenting on this is a good addition for the discussion section.

Discussion section:

Did other modeling methods (for instance the theoretical models described in your introduction) exhibit the speed-accuracy trade-off? How do other results of your model compare to the results of those models? Such a discussion on how your work is placed in the mathematical modeling world is still missing.

Methods section:

I understand now that slip-bonds dynamics are part of both equation 1 and 2. I do not really understand the different between the two. I understand why it is part of equation 2. But, for equation 1, when the arm is retracting, the friction depends on slip-bond adhesion behavior. This part I do not understand. I would expect an opposing force due to adhesion given by the variable n, and where n depends on slip-bond dynamics. I must be missing something. Can the authors please explain this better in the Methods section?

Line 494. I would expect the explanation and rationale of which parameters you change to simulate the effect of these drugs to be in the main text and not in the Appendix. Also mention the spcific drugs CK666 and blebbistatin in the main text.

Minor comments:

Second line of abstract. “Further complicated” doesn’t sound like the right term to me. It suggests that chemical signals also complicate cell motion. Then also, the geometrical aspect is not part of the abstract after this sentence.

Last sentence of abstract: “at the expense of compromising their” choose one the two

line 14: chamotaxis typo

line 91: the heaviside function doesn’t appear in the formulas anymore

Line 116: explain that c is an inhibitory polarity cue

In response to my previous questions you said that “this load is shared equally among n_i

bonds”. Please mention this assumption in Line 100-102.

Line 432/433: there is an empty red box in the white space at the left of this paragraph, that I can click and it leads me to page 20

Line 520-527: There is some repetition here

equation S11 in Appendix S1: some exp are italic

Line 495: there is a small extra red box above the red box around Appendix.

Reviewer #3: I thank the authors for a careful revision of the paper. While I am in agreement with a majority of revisions made by the authors, I have a couple of major concerns which I believe are absolutely crucial to this paper, and must be addressed before I can recommend publication.

(i) The authors mention: “We note that a recent study found faster cells to have lower chemotactic index [42], where motility was tuned using temperature rather than drug treatment.” I checked this reference, this paper does not make this claim at all, in fact the discussion in this paper says this: “cell migration speed increased with temperature (Figure 1F), without significant changes in the chemotactic index (CI) (Figure 1G)”. The data presented in this paper does not provide any clear evidence to show that faster cells do worse chemotaxis, and moreover, the authors of ref 42 do not make any claims about chemotactic index. Therefore, I would strongly recommend removing this part entirely.

(ii) “we reach the conclusion that the neutrophils correspond to the high beta regime of our model”, this is very vague – you cannot just map one particular cell to a certain parameter without any point of reference (conceptually this is like saying trains have a high speed, but with respect to what?), it would make sense to conclude that neutrophils have a higher beta than some other cell type for example. I believe that the authors should comment only on the qualitative correlation between the model and experiments, and not make any quantitative statements regarding the experiments, because there isn’t any quantitative similarity.

In the abstract the authors write: “We find that the model captures the details of the subcellular response to the chemokine gradient, as well as the large scale migration, suggesting that the neutrophils behave as fast cells, at the expense of compromising their chemotaxis accuracy, which explains the functionality of these immune cells.” I find this statement in the abstract to be misleading because the authors do not find any proof of the speed accuracy trade-off. If the authors want to stick to the comparison with experiments in this paper, I think they should restrict themselves to commenting only on the fact that the chemotaxis predicted by the model has qualitative similarity with neutrophil migration seen in experiments, thereby providing a mathematical framework to study chemotaxis in branched networks. Statements like a model capturing a detailed cell migration response can only be made if there is experimental proof for the model prediction (speed accuracy tradeoff in this case). Therefore, I would suggest to modify the abstract accordingly.

**Have the authors made all data and (if applicable) computational code underlying the findings in their manuscript fully available?**

Reviewer #1: Yes

Reviewer #2: Yes

Reviewer #3: Yes

PLOS authors have the option to publish the peer review history of their article (what does this mean?). If published, this will include your full peer review and any attached files.

Reviewer #1: No

Reviewer #2: **Yes:** Elisabeth Geraldine Rens

Reviewer #3: No

**Figure resubmission:**
---

## [Editor Report · Decision Letter 2]

21 Jan 2026

Dear Liu,

We are pleased to inform you that your manuscript 'Modelling chemotaxis of branched cells in complex environments provides insights into immune cell navigation' has been provisionally accepted for publication in PLOS Computational Biology.

Best regards,

Calina Copos, Ph.D.

Academic Editor

PLOS Computational Biology

Dimitrios Vavylonis

Section Editor

PLOS Computational Biology

---

## [Editor Report · Acceptance letter]

PCOMPBIOL-D-25-01385R2

Modelling chemotaxis of branched cells in complex environments provides insights into immune cell navigation

Dear Dr Liu,

I am pleased to inform you that your manuscript has been formally accepted for publication in PLOS Computational Biology. Your manuscript is now with our production department and you will be notified of the publication date in due course.

With kind regards,

Anita Estes
